# Silencing cryptic specialized metabolism in *Streptomyces* by the nucleoid-associated protein Lsr2

Emma J Gehrke[1,2], Xiafei Zhang[1,2], Sheila M Pimentel-Elardo[3], Andrew R Johnson[4], Christiaan A Rees[5], Stephanie E Jones[1,2], Hindra[1], Sebastian S Gehrke[2,6], Sonya Turvey[1,2], Suzanne Boursalie[1,2], Jane E Hill[5], Erin E Carlson[4,7], Justin R Nodwell[3], Marie A Elliot[1,2]*

[1]Department of Biology, McMaster University, Hamilton, Canada; [2]Michael G. DeGroote Institute for Infectious Disease Research, McMaster University, Hamilton, Canada; [3]Department of Biochemistry, University of Toronto, Toronto, Canada; [4]Department of Chemistry, Indiana University, Bloomington, United States; [5]Geisel School of Medicine and Thayer School of Engineering, Dartmouth College, Hanover, United States; [6]Department of Biochemistry and Biomedical Sciences, McMaster University, Hamilton, Canada; [7]Department of Chemistry, University of Minnesota, Minneapolis, United States

*For correspondence:
melliot@mcmaster.ca

**Abstract** Lsr2 is a nucleoid-associated protein conserved throughout the actinobacteria, including the antibiotic-producing *Streptomyces*. *Streptomyces* species encode paralogous Lsr2 proteins (Lsr2 and Lsr2-like, or LsrL), and we show here that of the two, Lsr2 has greater functional significance. We found that Lsr2 binds AT-rich sequences throughout the chromosome, and broadly represses gene expression. Strikingly, specialized metabolic clusters were over-represented amongst its targets, and the cryptic nature of many of these clusters appears to stem from Lsr2-mediated repression. Manipulating Lsr2 activity in model species and uncharacterized isolates resulted in the production of new metabolites not seen in wild type strains. Our results suggest that the transcriptional silencing of biosynthetic clusters by Lsr2 may protect *Streptomyces* from the inappropriate expression of specialized metabolites, and provide global control over *Streptomyces'* arsenal of signaling and antagonistic compounds.
DOI: https://doi.org/10.7554/eLife.47691.001

## Introduction

Chromosomes are remarkably dynamic molecules. In eukaryotes, chromosome structure is governed largely by histones, while in bacteria, organization is provided by the nucleoid-associated proteins. Collectively, these proteins function both in architectural capacities and in regulatory roles. Chromosome evolution in bacteria can be driven by mutation, genome rearrangement, and horizontal gene transfer, and work over the last decade has revealed that many bacteria have co-opted nucleoid-associated proteins to additionally serve as 'genome sentinels', suppressing the inappropriate expression of newly acquired DNA (*Dorman, 2007*; *Dorman, 2014*). This is thought to maximize competitive fitness by repressing the expression of foreign DNA until it is either incorporated into the existing regulatory networks of the host, or decays to a point that it is lost from the chromosome (*Navarre et al., 2007*).

Different bacteria employ distinct nucleoid-associated proteins as xenogeneic silencers, including H-NS in the proteobacteria, MvaT/MvaU in the pseudomonads (*Castang et al., 2008*), Rok in

*Bacillus* species (*Smits and Grossman, 2010*), and Lsr2 in the actinobacteria (*Gordon et al., 2008*). None of these proteins share sequence or structural homology, but all act by binding to AT-rich regions within the chromosome (*Navarre et al., 2006*; *Castang et al., 2008*; *Gordon et al., 2010*; *Smits and Grossman, 2010*). H-NS has been the best-studied of these proteins. In *Escherichia coli* and *Salmonella,* H-NS represses the expression of pathogenicity islands, endogenous phage genes, as well as other genes needed to respond to environmental changes (*Lucchini et al., 2006*; *Navarre et al., 2006*). H-NS binds DNA as a dimer, and can either polymerize along the DNA to form a rigid filament (*Liu et al., 2010*), or bridge DNA to facilitate chromosome compaction (*Dame et al., 2000*; *Dame et al., 2006*); both activities can limit the activity of RNA polymerase. Lsr2 is thought to function similarly to H-NS. To date, its study has been confined to the mycobacteria, where Lsr2 specifically binds and represses the expression of horizontally transferred genomic islands and AT-rich regions, including major virulence factor-encoding genes (*Gordon et al., 2010*).

In contrast to many of the pathogens in which chromosome organization and genome silencing have been explored, the streptomycetes are largely benign, sporulating soil bacteria (*Flärdh and Buttner, 2009*) that are instead renowned for their ability to produce a wide array of specialized metabolites (*Hopwood, 2007*; *Barka et al., 2016*). Notably, the metabolic output of this actinobacterial genus includes the majority of naturally-derived antibiotics used to treat bacterial infections. The streptomycetes encode two Lsr2 paralogs, unlike their mycobacterial relatives who possess a single *lsr2* gene. *Streptomyces* are additionally unusual in that they have linear chromosomes, where the majority of the genes required for viability are clustered in the chromosome core, and more species-specific and laterally-acquired genes are located in the flanking chromosome arms (*Bentley et al., 2002*). It is within these arm regions that most of the specialized metabolic clusters are found. Recent work has revealed that specialized metabolic clusters are over-represented as horizontally-transferred elements in the streptomycetes (*McDonald and Currie, 2017*), and that in the closely-related *Salinospora*, lateral gene transfer is a major driver of specialized metabolism (*Ziemert et al., 2014*).

Specialized metabolic gene clusters are subject to complex, hierarchical regulatory control (*van Wezel and McDowall, 2011*; *Liu et al., 2013*). Most *Streptomyces* clusters contain dedicated pathway-specific regulators, which in turn are controlled by a suite of more globally-acting transcription factors. Interestingly, however, most clusters are poorly expressed under normal laboratory conditions, and in many cases their associated metabolites remain uncharacterized. This is also the case for the filamentous fungi, many of whom have a broad, untapped specialized metabolic repertoire, courtesy of transcriptional silencing by histones (*Pfannenstiel and Keller, 2019*). Significant efforts are being made to stimulate the production of these 'cryptic' metabolites in both bacteria and fungi, as they are widely regarded as productive sources of new natural products (*Craney et al., 2013*; *Ochi and Hosaka, 2013*; *Scharf and Brakhage, 2013*; *Yoon and Nodwell, 2014*; *Daniel-Ivad et al., 2017*; *Onaka, 2017*).

We sought to investigate the role of the nucleoid-associated proteins Lsr2 and LsrL in gene regulation in *Streptomyces*. We found that deleting *lsr2* from the chromosome of *Streptomyces venezuelae* had minor effects on *S. venezuelae* growth and development and major effects on metabolism. In contrast, deleting *lsrL* had no detectable impact on development, and only a minor effect on metabolism. Focussing on Lsr2, we determined that it bound AT-rich regions, generally repressed the expression of prophage genes and other genes unique to *S. venezuelae* (presumably acquired by lateral gene transfer), and suppressed antisense gene expression. The most profound effect of *lsr2* deletion, however, was the large-scale activation of specialized metabolic cluster gene expression. Lsr2 directly repressed the transcription of many cryptic clusters in a way that is analogous to Lsr2- and H-NS-mediated repression of pathogenicity islands in other bacteria, and histone-mediated cluster silencing in fungi. Unexpectedly, Lsr2 also controlled the expression of well-characterized and highly-conserved clusters, suggesting that Lsr2 control has been broadly integrated into the regulatory cascades governing specialized metabolism. Our results suggest that Lsr2 functions as a metabolic gatekeeper in the streptomycetes, playing a critical role in the metabolic circuitry of these organisms, and that bacteria, like fungi, employ chromosome structuring elements to control specialized metabolism. Finally, we have manipulated Lsr2 activity using dominant negative variants, and successfully promoted the production of otherwise cryptic metabolites in a wide range of *Streptomyces* species.

## Results

### Lsr2 and Lsr2-like (LsrL) in *Streptomyces venezuelae*

All *Streptomyces* species possess two *lsr2* homologs (*Chandra and Chater, 2014*). We examined these gene products and found that within any given species, the homologs shared ~50% end-to-end amino acid identity and 60–65% sequence similarity. One homolog shared both genomic synteny and greater sequence similarity when compared with Lsr2 from *M. tuberculosis* (*Figure 1—figure supplement 1*). We termed this protein Lsr2 (SVEN_3225 in *S. venezuelae*), and its more divergent homolog LsrL, for Lsr2-like (SVEN_3832 in *S. venezuelae*).

To determine the biological role of these two genes and their products, we constructed single and double mutants in *S. venezuelae*. Deleting *lsr2* had no observable effect on *S. venezuelae* growth in liquid culture (*Figure 1A*). Sporulation was, however, reproducibly delayed in the mutant (*Figure 1A,B*), and the 'exploration' capabilities of the mutant (*Jones et al., 2017*) were altered compared with wild type (*Figure 1—figure supplement 2*). Metabolism was also affected, with the mutant strain producing enhanced levels of melanin compared with its wild type parent (*Figure 1B*). Melanin over-production and the delay in sporulation could be partially complemented through the in trans expression of *lsr2* (*Figure 1—figure supplement 2*). In contrast, deleting *lsrL* had no discernable effect on *S. venezuelae* growth or development, while a double *lsr2 lsrL* mutant strain most closely resembled the *lsr2* single mutant (*Figure 1*).

We examined the expression of *lsr2* and *lsrL* throughout the *S. venezuelae* life cycle using RNA sequencing data collected at three time points, and found *lsr2* transcripts reached maximal levels during the later stages of liquid culture growth (*Figure 1C*). In comparison, *lsrL* levels were lower and peaked earlier (*Figure 1C*). We focussed our subsequent investigations on Lsr2, given its higher transcript levels and more pronounced mutant phenotype.

### Lsr2 represses the expression of horizontally acquired and specialized metabolic genes in *S. venezuelae*

To begin understanding the effect of Lsr2 on *Streptomyces* growth and metabolism, we isolated RNA samples from wild type and *lsr2* mutant strains (see above), and compared transcript levels for the two strains using RNA-sequencing. The most striking differences were seen at the final growth stage (third time point), coinciding with the timing of maximal *lsr2* expression (*Figure 1C*). Using a stringent cut-off (>4 fold change, with a *q* value < 0.01), we found that 484 genes has significantly altered expression in the *lsr2* mutant relative to wild type (*Supplementary file 1*). This represented ~6% of genes in the chromosome, and for the vast majority (>90%) of these genes, expression was up-regulated. These differentially expressed genes included many horizontally acquired genes: ~10% were phage-related (43 of 484), and an additional 10% (47 genes) were unique to *S. venezuelae*, relative to the other *Streptomyces* genomes available in StrepDB (strepdb.strepto-myces.org.uk/) (*Figure 2A*; *Supplementary file 1*). We also observed increased antisense expression in a number of instances, suggesting that like its H-NS counterpart in *E. coli*, Lsr2 can also suppress intragenic transcription (*Singh et al., 2014*) (*Figure 2—figure supplement 1*).

We noted that *lsrL* expression was significantly increased relative to wild type (*Supplementary file 1*). There was, however, no significant change in the expression of any of the well-characterized developmental genes in the *lsr2* mutant (*Supplementary file 1*). In contrast, specialized metabolic genes were significantly and disproportionately affected by Lsr2, with 155 of 484 differentially impacted genes (*Supplementary file 1*) localized to specialized metabolic clusters; this corresponded to ~15% of all predicted specialized metabolic genes in the chromosome, compared with altered expression for <5% of all others (non-specialized metabolic genes) (binomial test, p<0.00001). Consistent with the enhanced brown pigmentation observed for the *lsr2* mutant, we observed increased expression for genes in one of the predicted melanin biosynthetic clusters (*Table 1*; *Supplementary file 1*). Overall, deleting *lsr2* led to altered expression for genes within 21 of 30 predicted or characterized specialized metabolic clusters, with increased expression (ranging from 4 to ~900×) observed for genes in 18 of these clusters (*Table 1*; *Supplementary file 1*). Indeed, for six clusters, more than one-third of their genes were significantly upregulated (*Table 1*; *Supplementary file 1*); predicted cluster boundaries are typically very generous (*e.g.* the chloramphenicol cluster is predicted to extend from *sven_0902–0945*, while the experimentally validated

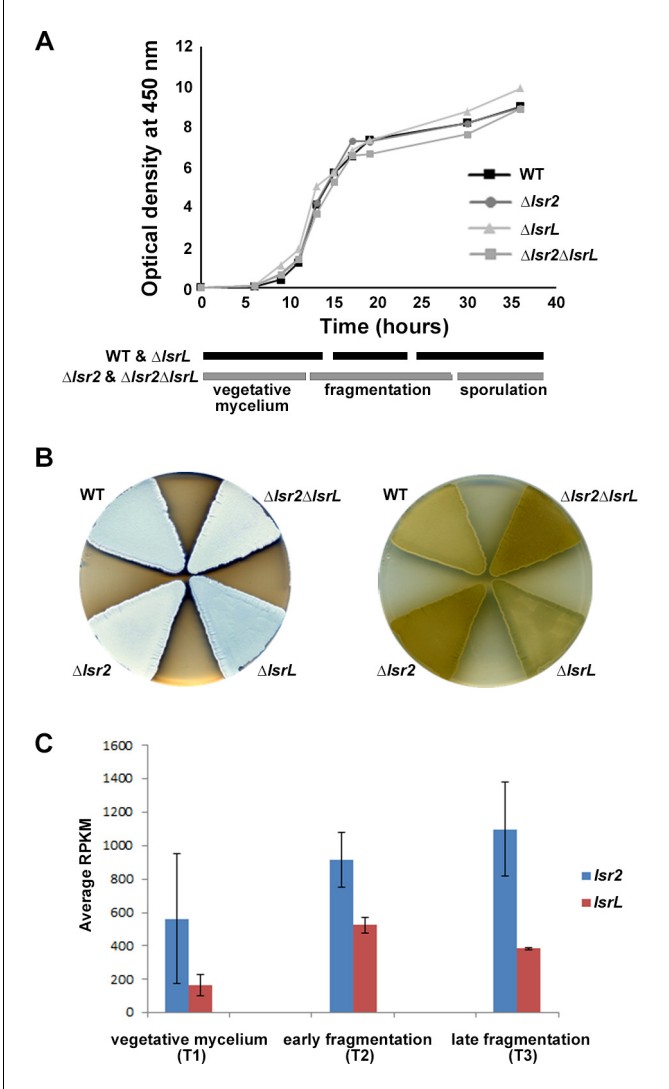

**Figure 1.** Expression and phenotypic analyses of *lsr2* and *lsrL* in *S. venezuelae*. (**A**) Growth curves and developmental stages (as determined using light microscopy) of wild type (WT), and the three different *lsr2/lsrL* mutants, over a 40 hr time course in MYM liquid medium. (**B**) Comparing development (left) and melanin/brown pigment production (right – underside of plate) of wild type (WT), versus single (Δ*lsr2* and Δ*lsrL*) and double (Δ*lsr2*Δ*lsrL*) mutant strains after 2 d growth on MYM agar medium. The white color of the *lsr2*-containing mutants reflects aerial hyphae formation, while the darker color associated with the wild type and *lsrL* mutant strains indicates culture sporulation. (**C**) Comparative transcript levels (RPKM = reads per kilobase per million) for *lsr2* and *lsrL* at three time points [T1, T2, T3, representing the three developmental stages indicated in (A)] in liquid MYM medium, as assessed using RNA-sequencing data. Data are presented as mean ± standard deviation ($n = 2$).
DOI: https://doi.org/10.7554/eLife.47691.002

The following figure supplements are available for figure 1:

**Figure supplement 1.** Sequence similarity and genetic organization of *lsr2* and *lsrL* (*lsr2*-like) genes and their products in *Streptomyces* and *Mycobacterium*.
DOI: https://doi.org/10.7554/eLife.47691.003

**Figure supplement 2.** Phenotypic comparison of *lsr2* deletion mutants with wild-type *S. venezuelae*.
DOI: https://doi.org/10.7554/eLife.47691.004

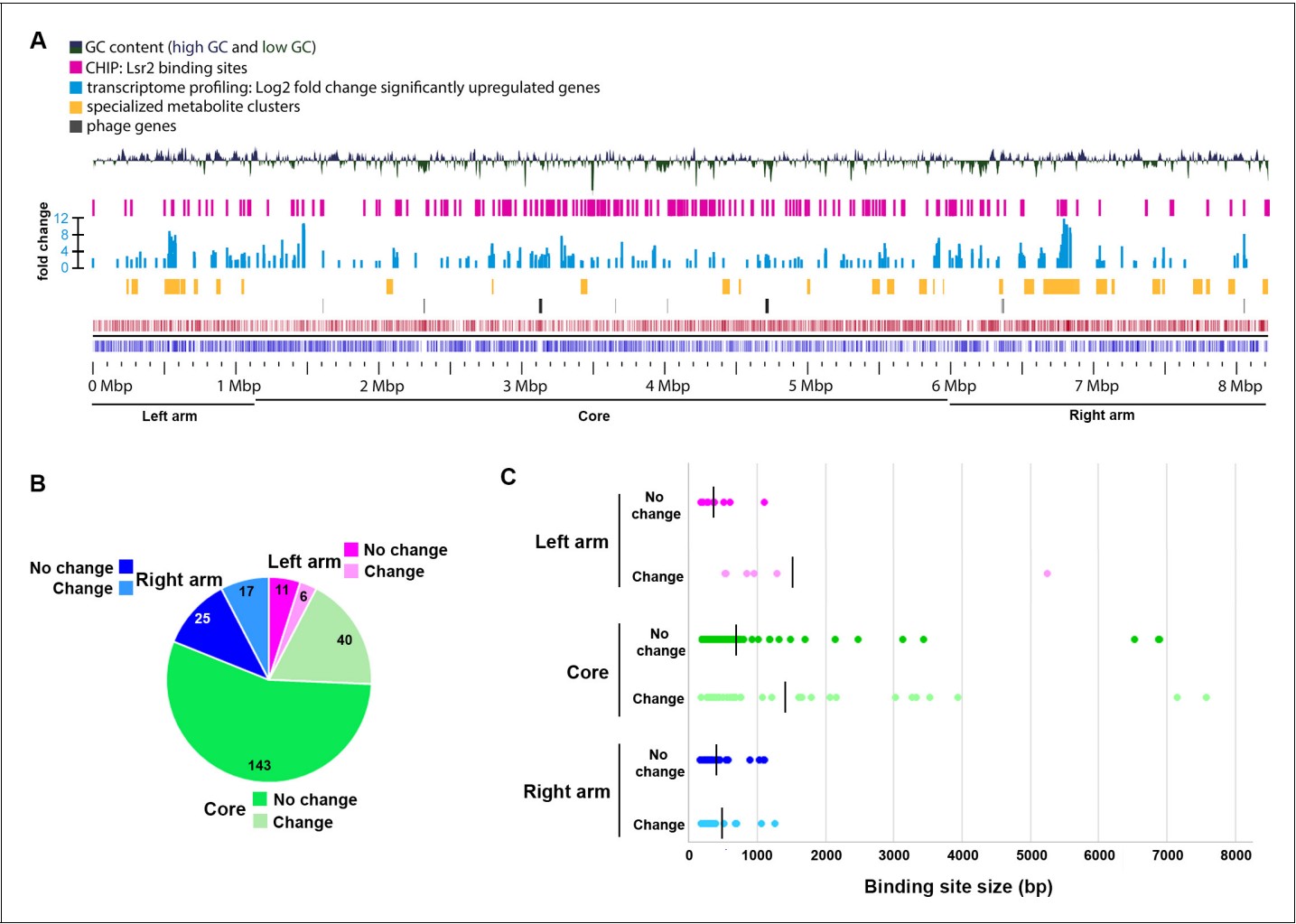

**Figure 2.** Composition of *S. venezuelae* chromosome in relation to Lsr2 binding sites and differentially affected genes. (A) Panels are described from the bottom up. Bottom panels: coding sequences and relative strand organization (forward orientation shown in red; reverse orientation shown in blue) across the *S. venezuelae* chromosome, with left arm, core and right arm regions indicated. Above that, the regions shown with black bars indicate the relative position of predicted phage genes, while those in yellow indicate the location of specialized metabolic clusters. The light blue bars represent genes whose expression is significantly upregulated in the *lsr2* mutant (fold change indicated), while the pink bars indicate Lsr2 binding sites, as determined by ChIP-seq. The top panel depicts the GC content of the chromosome, relative to the average percentage (72.4%). The peaks above the middle line indicate a GC content above 72.4%, while those below indicate a GC content below 72.4%. Image was generated using GView (*Petkau et al., 2010*). (B) Binding sites within the left arm (pink), core (green) or right arm (blue), and relative proportion of sites associated with transcriptional changes (no change in transcription = darker color; change in transcription = lighter color). Shown within each segment is the number of binding sites associated with that region. (C) Size of binding sites (in base pairs, bp) in each of the chromosome regions, separated into those associated with altered transcription (change), versus not (no change). Color scheme is as described in (B). The average binding site size for each group is indicated by the vertical black line.

DOI: https://doi.org/10.7554/eLife.47691.005

The following figure supplement is available for figure 2:

**Figure supplement 1.** Increased antisense RNA expression in an *lsr2* mutant strain compared with wild type.

DOI: https://doi.org/10.7554/eLife.47691.006

cluster encompasses *sven_0913–0928*), and as a result we anticipate that the relevant proportion of upregulated genes in these clusters is actually much higher. Importantly, of these six clusters, five were largely transcriptionally silent in the wild type strain (*Figure 3*), with an average RPKM for the differentially affected genes being <10 in the wild type, compared with an average of >190 in the *lsr2* mutant (*Supplementary file 1*).

**Table 1.** Specialized metabolic clusters and their control by Lsr2.

| Predicted specialized metabolic gene cluster[#] (characterized product) | 1st gene | Last gene | Number of upregulated genes (q-value < 0.01; >4 fold change) | % of upregulated genes in each cluster | Number of Lsr2-associated sites (q-value < 0.01) |
|---|---|---|---|---|---|
| Ectoine | SVEN_0223 | SVEN_0234 | 1 | 8.33 | 0 |
| Terpene | SVEN_0261 | SVEN_0306 | 1 | 0 | 0 |
| **T1PKS-T3PKS-NRPS** (*venemycin/watasemycin/thiazostatin*) | **SVEN_0463** | **SVEN_0531** | 28 | 42.65 | 2 (SVEN_0502*, 506) |
| Lantipeptide – terpene | SVEN_0540 | SVEN_0561 | 0 | 0 | 1 (SVEN_0557) |
| Lantipeptide (*venezuelin*) | SVEN_0612 | SVEN_0630 | 3 (1 repressed) | 16.67 | 0 |
| Indole (*acryriaflavin*) | SVEN_0755 | SVEN_0772 | 0 | 0 | 0 |
| **Chloramphenicol** | **SVEN_0913** | **SVEN_0928** | 14 | 93.33 | 1 (SVEN_0926) |
| Other | SVEN_1844 | SVEN_1884 | 0 | 0 | 0 |
| Siderophore (*desferrioxamine-like*) | SVEN_2566 | SVEN_2577 | 5 | 41.67 | 1 (SVEN_2576) |
| Lassopeptide | SVEN_3103 | SVEN_3132 | 2 | 6.90 | 1 (SVEN_3116*−7) |
| Other | SVEN_4061 | SVEN_4110 | 1 | 2.04 | 1 (SVEN_4069–70) |
| Butyrolactone (*gaburedin*) | SVEN_4179 | SVEN_4189 | 0 | 0 | 0 |
| **Melanin** | **SVEN_4620** | **SVEN_4662** | 0 | 0 | 4 (SVEN_4629–30, 4632, 4634–5, 4651) |
| Butyrolactone | SVEN_5076 | SVEN_5111 | 3 | 11.54 | 2 (SVEN_5091–92, 5106*−07) |
| Thiopeptide | SVEN_5119 | SVEN_5145 | 3 | 11.54 | 3 (SVEN_5127–8, 5129–31, 5132–3) |
| T3pks | SVEN_5351 | SVEN_5383 | 0 | 0 | 0 |
| Siderophore | SVEN_5413 | SVEN_5426 | 0 | 0 | 0 |
| Siderophore | SVEN_5471 | SVEN_5482 | (1 repressed) | 9.09 | 0 |
| **Bacteriocin** | **SVEN_5817** | **SVEN_5840** | 3 | 15.15 | 1 (SVEN_5817*) |
| **Butyrolactone – T2PKS** | **SVEN_5951** | **SVEN_6002** | 19 | 45.10 | 6 (SVEN_5963*−4, 5968*−9, 5972–3, 5974, 5975–6, 5979) |
| Other | SVEN_6112 | SVEN_6204 | 3 (1 repressed) | 3.26 | 1 (SVEN_6199) |
| NRPS-ladderane | SVEN_6134 | SVEN_6282 | 50 | 36.49 | 7 (SVEN_6199, 6216–7, 6219–20, 6225*, 6230, 6247–8, 6251) |
| Terpene | SVEN_6436 | SVEN_6490 | 6 | 16.67 | 1 (SVEN_6458) |
| Bacteriocin | SVEN_6527 | SVEN_6535 | (1 repressed) | 0 | 0 |
| T2PKS | SVEN_6767 | SVEN_6814 | 2 | 4.26 | 0 |
| Melanin | SVEN_6833 | SVEN_6842 | 4 | 44.44 | 0 |
| NRPS | SVEN_7032 | SVEN_7080 | 0 | 0 | 0 |
| **Terpene** | **SVEN_7101** | **SVEN_7119** | (two repressed) | 0 | 1 (SVEN_7109–10*) |
| **T3PKS** | **SVEN_7223** | **SVEN_7259** | 1 | 2.78 | 2 (SVEN_7235, 7237*−8) |
| **Terpene-NRPS** | **SVEN_7417** | **SVEN_7452** | 0 | 0 | 4 (SVEN_7427–8, 7440, 7447–9, 7449–50) |

[#]:Cluster prediction by antiSMASH; * Asterisks indicate regulatory genes bound by Lsr2

Bold: clusters containing an Lsr2 binding site

Gray shading: clusters containing differentially expressed genes

DOI: https://doi.org/10.7554/eLife.47691.012

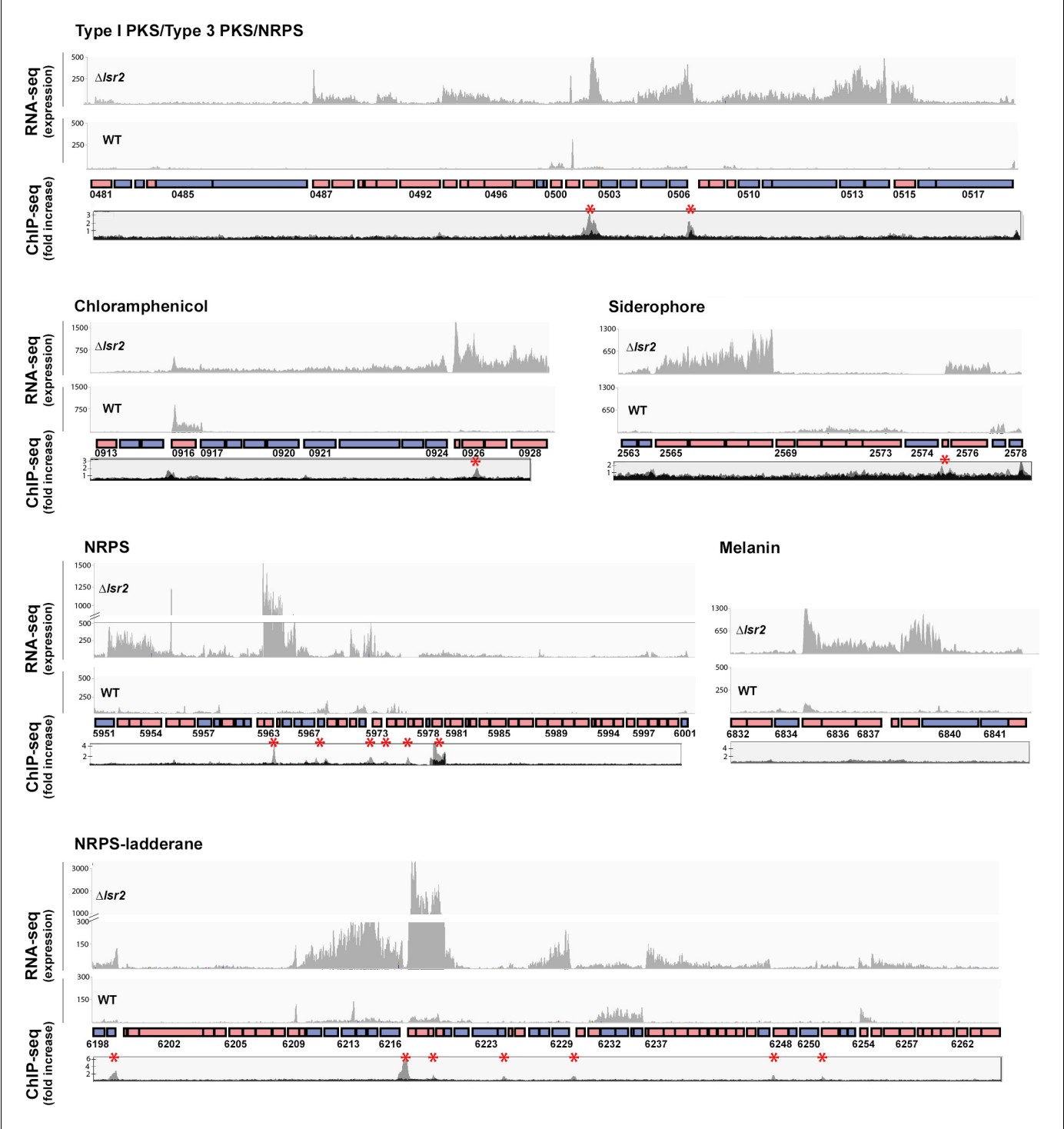

**Figure 3.** Lsr2 binding sites and expression analysis of select Lsr2-regulated specialized metabolic clusters. For each of the six specialized metabolic clusters shown, genes oriented in the forward direction are shown as pink boxes, while those in the reverse direction are shown as blue boxes. Select genes are labeled with their corresponding *sven* numbers (*e.g.* 0481). RNA-sequencing results are shown above each cluster, with graphs depicting expression levels. For each, the *lsr2* read profile is shown on the top, while the wild type profile is shown on the bottom. The ChIP-seq profiles (below the gene cluster) are shown as 'fold increase', with the gray profiles indicating regions associated with 3× FLAG tagged Lsr2 (where an anti-FLAG antibody was used for the immunoprecipitation), and the black profile representing the negative control (strain expressing a non-FLAG-tagged Lsr2 variant). Lsr2−3× FLAG binding sites are indicated with a red asterisk.

DOI: https://doi.org/10.7554/eLife.47691.007

*Figure 3 continued on next page*

*Figure 3 continued*

The following figure supplements are available for figure 3:

**Figure supplement 1.** Semi-quantitative RT-PCR using RNA (biological replicates) isolated from *S. venezuelae* wild type (WT) or Δ*lsr2* in early stationary phase.

DOI: https://doi.org/10.7554/eLife.47691.008

**Figure supplement 2.** Electrophoretic mobility shift assays with Lsr2.

DOI: https://doi.org/10.7554/eLife.47691.009

**Figure supplement 3.** Schematic of the Lsr2-binding site between *sven_5106–5107*.

DOI: https://doi.org/10.7554/eLife.47691.010

**Figure supplement 4.** Effect of increasing GC content on Lsr2 binding and activity.

DOI: https://doi.org/10.7554/eLife.47691.011

We experimentally validated our RNA sequencing results using reverse transcription-PCR, focussing on select genes from clusters specifying a range of predicted products (polyketides, thiopeptides, butyrolactones, and non-ribosomal peptides). In each case, transcripts were only reproducibly detected in the *lsr2* null mutant (*Figure 3—figure supplement 1*). This suggested that Lsr2 functions as a principal regulator for the majority of specialized metabolites in *S. venezuelae*, repressing the activity of these clusters under laboratory conditions.

## Lsr2 directly controls specialized metabolite cluster expression

To determine whether the effects observed in our transcription profiling experiments stemmed from direct or indirect control by Lsr2, we examined our RNA-seq data for significant changes in the expression of any known global antibiotic regulators. Unexpectedly, none were affected by the loss of Lsr2 (*Supplementary file 2*), suggesting that Lsr2 may directly impact specialized metabolic cluster expression. We next conducted chromatin-immunoprecipitation (ChIP) sequencing experiments to identify Lsr2-associated DNA sequences using a functional Lsr2−3× FLAG fusion-expressing strain (*Figure 1—figure supplement 2*). We isolated immunoprecipitated DNA from FLAG-tagged and control (untagged Lsr2-expressing) strains late during liquid culture growth, at a developmental stage corresponding to the third time point in our transcriptional analyses. Using a stringent filter (*q* value < 0.01), we identified 223 Lsr2 binding sites distributed throughout the chromosome (*Figure 2A*; *Supplementary file 3*). These included sites within 17 specialized metabolic clusters (*Table 1*), with 14 of these clusters showing altered transcriptional profiles in the *lsr2* mutant compared with the wild type strain (*e.g. Figure 3*).

The effect of Lsr2 on specialized metabolism, and the lack of association with other characterized global regulators, collectively suggested that Lsr2 activity may represent a new level in the regulatory cascades governing specialized metabolism. As most globally acting antibiotic regulators exert their effects by controlling the expression of pathway-specific regulators (*e.g. McKenzie and Nodwell, 2007*; *Rigali et al., 2008*; *Gao et al., 2012*; *Wang et al., 2013*), we tested whether this was also the case for Lsr2. For the 14 clusters that were bound directly by Lsr2 and had altered transcription profiles, we examined where Lsr2 bound, relative to any potential cluster-specific regulators. For approximately half (8 of 14), Lsr2 binding was associated with a regulatory gene (*Table 1*). For the others, Lsr2 bound elsewhere in the cluster, suggesting an independent mechanism of regulation.

## Trends in Lsr2 binding and regulatory control

To better understand how Lsr2 exerted its repressive effects, we undertook a more comprehensive investigation into its binding and regulatory impacts. We first validated the specificity of Lsr2 binding using electrophoretic mobility shift assays (EMSAs). We tested five ChIP-associated sequences, and found that each of these effectively out-competed non-specific DNA probes for binding by Lsr2. This indicated that Lsr2 preferentially bound the DNA regions identified in our ChIP assays (*Figure 3—figure supplement 2*).

Our ChIP-seq results suggested that Lsr2 bound to 223 sites across the chromosome. Interestingly, these sites were not concentrated in the arm regions where more of the species-specific (and presumably laterally-acquired) sequences were located, but instead were enriched in the 'core' region of the chromosome (as defined by *Bentley et al., 2002*) (*Figure 2B*; *Supplementary file 3*).

When considering all Lsr2 binding sites,~25% of the associated genes (where Lsr2 bound immediately upstream and/or overlapping with their coding sequences) had altered transcriptional profiles, and of these, more than 30% were in specialized metabolic clusters (19 of 63) (*Table 1*; *Supplementary file 3*).

We assessed whether there was any correlation between binding site position and regulatory impact. We found that binding sites within the arm regions were more likely to have transcriptional ramifications compared with those in the core [35% (left arm), 40% (right arm), 25% (core)] (*Figure 2B*). Binding sites associated with transcriptional changes were also, on average, larger than those that had no direct effect on transcription, at least for the left and core regions (*Supplementary file 3*; *Figure 2C*), although there were a number of large sites within the core region that had no direct effect on transcription.

We next sought to understand whether there was any specificity to Lsr2 binding in the chromosome. We analyzed the in vitro-confirmed Lsr2 binding sites (from *Figure 3—figure supplement 2*) using the MEME server (*Bailey et al., 2009*); however, no consensus motif could be identified. In examining the cluster-associated binding sites more broadly, we found these sequences had an average GC-content of 62.9%. When all Lsr2 binding sites were considered, an average GC-content of 65% was observed (*Supplementary file 3*), well below the chromosome average of 72.4%.

Previous in vitro analyses of binding preferences for Lsr2 from *M. tuberculosis,* had defined an eight nucleotide AT-rich sequence as being optimal (*Gordon et al., 2011*). We analyzed both the *S. venezuelae* genome, and our identified Lsr2 binding sites, for either AT-rich 20 nt segments (≥50% A/T), or AT-rich 'core' sequences (defined as 5 of 6 consecutive nucleotides being A/T). To first determine the relative AT density in the *S. venezuelae* chromosome, we assessed the number of 20 nt AT-rich stretches in 30 randomly selected sequences – 15 that were 500 bp and 15 that were 1000 bp in length (*Supplementary file 4*). We found that 7/15 of the shorter sequences lacked any AT-rich stretch (with the number of stretches ranging from 0 to 20, with an average of 5), compared with 2/15 of the longer sequences (with numbers ranging from 0 to 36, and an average of 10). In contrast, the vast majority (222/223) of Lsr2 target sequences possessed at least one AT-rich 20 nt stretch: shorter target sequences (≤500 bp) contained anywhere from 0 to 27 non-overlapping stretches (average of 7), while longer sequences (≥750 bp) contained between 8 and 291 (average of 64) (*Supplementary file 4*).

We next assessed the presence of AT-rich core sequences within both the random genome sequences, and the Lsr2-bound sites (*Supplementary file 4*). For the random segments, 11/15 of the 500 bp sequences lacked an AT-rich core (with numbers ranging from 0 to 10, with an average <1). This closely mirrored the absence of an AT-rich core in 10/15 of the 1000 bp sequences (range of 0–8, with an average of 1.5). This is in stark contrast to the Lsr2 target binding sequences: only 9 of 223 target sequences lacked an AT-rich core, with shorter sequences (≤500 bp) averaging three core sites (ranging from 0 to 11), and larger sequences (≥750 bp) averaging 25 (ranging from 1 to 124). This collectively suggested that while the presence of an AT-rich core sequence and multiple AT-rich segments may not be sufficient to promote Lsr2 binding, they appear to be near universal characteristics of Lsr2-bound sequences.

To experimentally assess the importance of these AT-rich sequences for Lsr2 binding, we focussed our attention on the Lsr2 binding site located in the intergenic region between the divergently expressed *sven_5106* and *sven_5107* genes within the predicted butyrolactone biosynthetic cluster (*Table 1*; *Figure 3—figure supplement 3*). Using EMSAs, we compared Lsr2 binding to the wild type sequence (58% GC), with binding to mutant sequences having increasing GC content (63%, 64% and 70%). Lsr2 bound the AT-rich sequences with much higher affinity than the more GC-rich sequences, with very little binding observed for the 70% GC-containing probe (*Figure 3—figure supplement 4*). Notably, there was little difference in binding seen for sequences in which an AT-rich core was disrupted (64% GC), versus when the overall AT-content was changed (63% GC) (*Figure 3—figure supplements 3* and *4*). To determine whether the Lsr2 preference for more AT-rich DNA was also observed in vivo, we introduced these altered sequences in place of the wild type, within a larger DNA fragment (spanning 9 kb and encompassing *sven_5105–07*) on an integrating plasmid vector, and introduced these variants into the Lsr2–3× FLAG expressing strain. We conducted ChIP experiments for each strain, and quantified the amount of DNA specific for this region using quantitative PCR. We observed far higher levels of the wild type sequence compared with any of the mutant sequences (>80% less), presumably reflecting greater Lsr2 affinity for the wild type,

AT-rich sequence (*Figure 3—figure supplement 4*). We also assessed the expression of the flanking genes in each case, and observed little expression from the wild type sequence, while increased expression was associated with the mutant sequences. This indicated that decreased binding by Lsr2 led to increased transcription of the flanking genes (*Figure 3—figure supplement 4*).

## Lsr2 activity is not specific for newly acquired biosynthetic clusters

Given that Lsr2 is predicted to be functionally equivalent to H-NS in *E. coli*, where H-NS inhibits the expression of laterally-acquired DNA sequences, we wanted to determine whether Lsr2 preferentially bound and repressed the expression of recently acquired (poorly conserved) biosynthetic clusters in *S. venezuelae*. To investigate cluster conservation, we subjected the sequence of each of the 30 *S. venezuelae* specialized metabolic clusters to BLAST searches. We focussed our attention on eight phylogenetically divergent *Streptomyces* species, alongside three strains of *S. venezuelae*, and assessed the conservation of each cluster within these streptomycetes (*Figure 4*; *Table 1*). As expected, we found that Lsr2 bound and repressed the expression of genes in many poorly conserved/recently acquired clusters. However, not all *S. venezuelae*-specific clusters were controlled by Lsr2, and several of the best conserved clusters (*e.g.* siderophore/desferrioxamine-encoding biosynthetic cluster and bacteriocin-encoding clusters) were under direct Lsr2 control (*Figure 4*; *Table 1*). This suggested that Lsr2 may function both as a silencer of newly acquired clusters, and as a central regulator within the hierarchy governing specialized metabolic cluster expression.

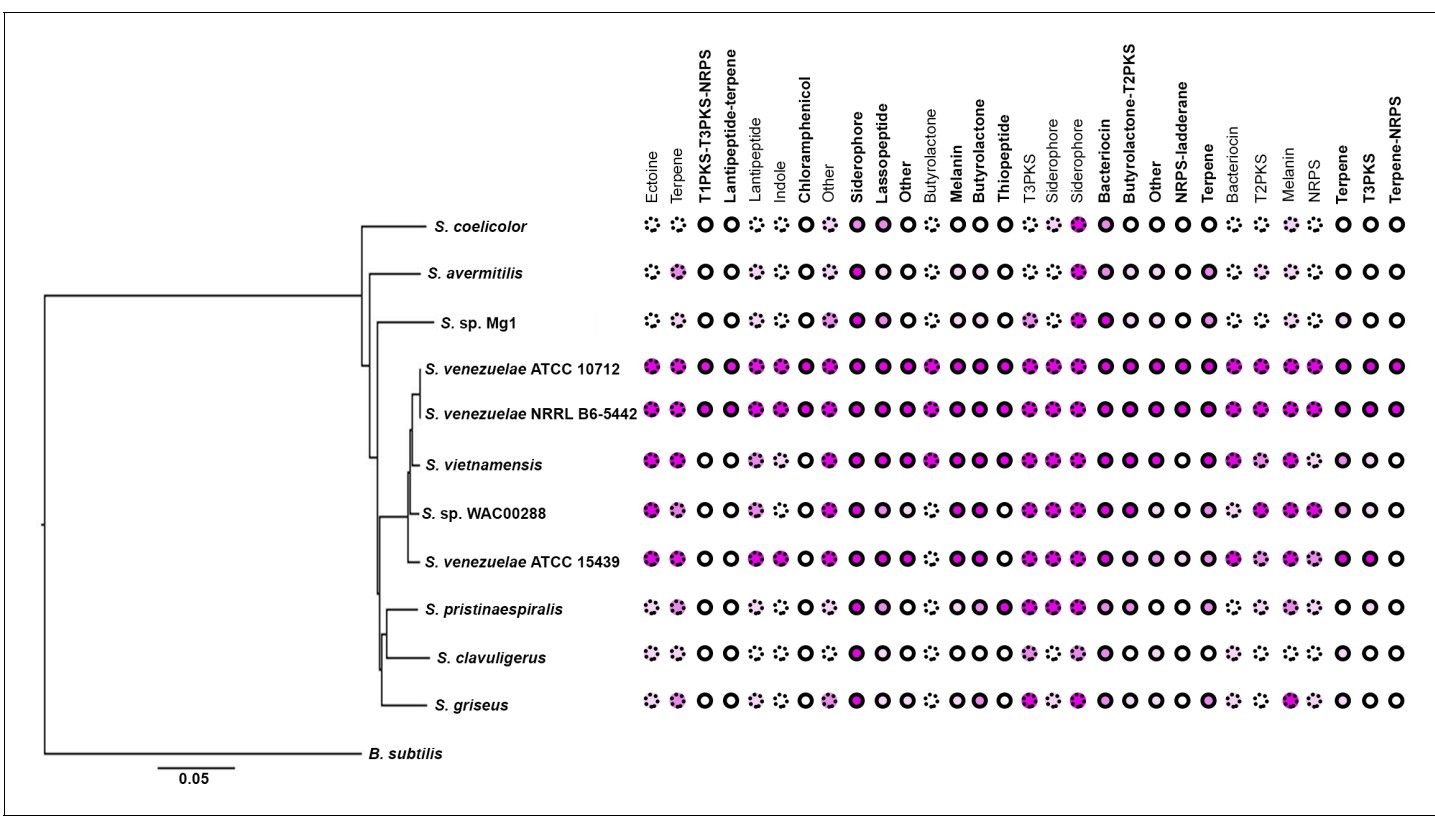

**Figure 4.** Conservation of specialized metabolic clusters in diverse streptomycetes. Phylogenetic tree of diverse *Streptomyces* species, with the relative conservation of each specialized metabolic cluster from *S. venezuelae* shown in the right. Conservation is based on BLAST analyses, with <20% (white), 20% (light pink), 40% (medium pink) and 60% (dark pink) indicating query coverage and overall degree of cluster conservation. Clusters bound by Lsr2 are indicated with bolded names and solid circles, while those not bound by Lsr2 are depicted with dotted circles.
DOI: https://doi.org/10.7554/eLife.47691.013

## Deleting Lsr2 reprograms specialized metabolism and yields novel compounds

Given the abundance of specialized metabolic genes affected by Lsr2, we examined the antibiotic production capabilities of the *lsr2* mutant. Crude methanol extracts from wild type and *lsr2* mutant cultures were initially tested against the Gram-positive indicator bacterium *Micrococcus luteus.* We observed a significant increase in growth inhibition for extracts from the *lsr2* mutant relative to the wild type strain (*Figure 5A*). Using activity guided fractionation and purification coupled with LC/MS analyses, we identified chloramphenicol as being the major inhibitory molecule (*Figure 5B*). Chloramphenicol is a well-known antibiotic, but it is not expressed at appreciable levels by *S. venezuelae* under normal laboratory conditions (*Figure 5B*) (*Fernández-Martínez et al., 2014*).

We next compared the soluble metabolites produced by wild type and *lsr2* mutant strains, and found each had a unique metabolic profile. We further tested the metabolic effects of deleting *lsrL,* and *lsr2* in conjunction with *lsrL,* as the increased *lsrL* expression observed in the *lsr2* mutant suggested that a double mutant may have more profound metabolic consequences than the *lsr2* mutation alone. Comparing the metabolic profiles of these four strains revealed that the greatest effect stemmed from the loss of *lsr2*, although the loss of *lsrL* (on its own, and in conjunction with *lsr2*) led to minor changes in metabolic output (*Figure 5—figure supplement 1*). In comparing the production of individual metabolites in a wild type and *lsr2* mutant strain, we first focussed our attention on compounds produced after 3 days of growth in liquid MYM medium. We observed unique peaks in the *lsr2* mutant for venemycin, a chlorinated venemycin derivative, as well as thiazostatin and watasemycin (*Figure 5C*). These compounds have all been described recently; however, this is the first time they have been shown to be produced in *S. venezuelae,* as their previous characterization required expression in a heterologous *Streptomyces* host (*Thanapipatsiri et al., 2016*; *Inahashi et al., 2017*).

Further examination of the soluble metabolites of 3, 4 and 5 day cultures grown in MYM medium yielded ESI(+) (electrospray ionization) metabolome profiles that were compared using XCMS, a tool to identify, quantify, and compare metabolite profiles across samples (*Smith et al., 2006*). While there were a multitude of new and enriched metabolites produced by the *lsr2* mutant, we focussed our attention on the most abundant compounds (intensities greater than $10^5$). Within these highly abundant metabolites, we identified six unique molecules produced only by the *lsr2* mutant (excluding isotopes, adducts, chemical noise and irreproducible peaks across replicates). An additional five compounds were significantly (>5×) more abundant in the *lsr2* mutant than wild type (*Supplementary file 5*). Of these new and enhanced compounds, only one was a known molecule (ferrioxamine), produced by a well-conserved cluster under Lsr2 control (*Table 1*).

Included amongst the unique compounds was a novel peak of *m/z* 281 in the *lsr2* mutant (*Figure 5D*). Based on fragmentation analysis, this compound was predicted to be *N*-acetyl-7-chloro-L-tryptophan. To determine the gene cluster responsible for producing this compound, we searched for halogenase-encoding genes. We identified *sven_6229* as a reasonable candidate, as it was dramatically (>200×) upregulated in an *lsr2* mutant (*Supplementary file 1*). It was also part of a large, otherwise transcriptionally silent specialized metabolic gene cluster (the 'NRPS-ladderane' cluster in *Figure 3* and *Table 1*). We mutated *sven_6229*, and found the *m/z* 281 peak disappeared (*Figure 5D*). A close homolog of SVEN_6229, PrnA, catalyzes the conversion of L-tryptophan to 7-chloro-L-tryptophan (*Dong et al., 2005*), and mutant PrnA variants can yield 5-chloro-L-tryptophan (*Lang et al., 2011*). We predicted that an acetylated form of one of these two chlorinated tryptophan molecules most likely corresponded to the novel *m/z* 281 molecule identified in the *lsr2* mutant. We synthesized the two analogs, and confirmed the identity of the unknown compound as being *N*-acetyl-chlorotryptophan, by virtue of the near identical MS/MS spectra of the synthetic candidates (*Figure 5—figure supplement 2*). Interestingly, however, co-elution studies with the synthetic standards clearly demonstrated that the unknown species was neither the 5-chloro, nor the 7-chloro isomer (*Figure 5—figure supplement 2*). We predict that this compound is a new tryptophan-derived precursor, that is likely incorporated into the *m/z* 668.2 compound that appeared at 4 days (*Supplementary file 5*), as this larger molecule also disappeared when *sven_6229* was mutated (*Figure 5—figure supplement 3*).

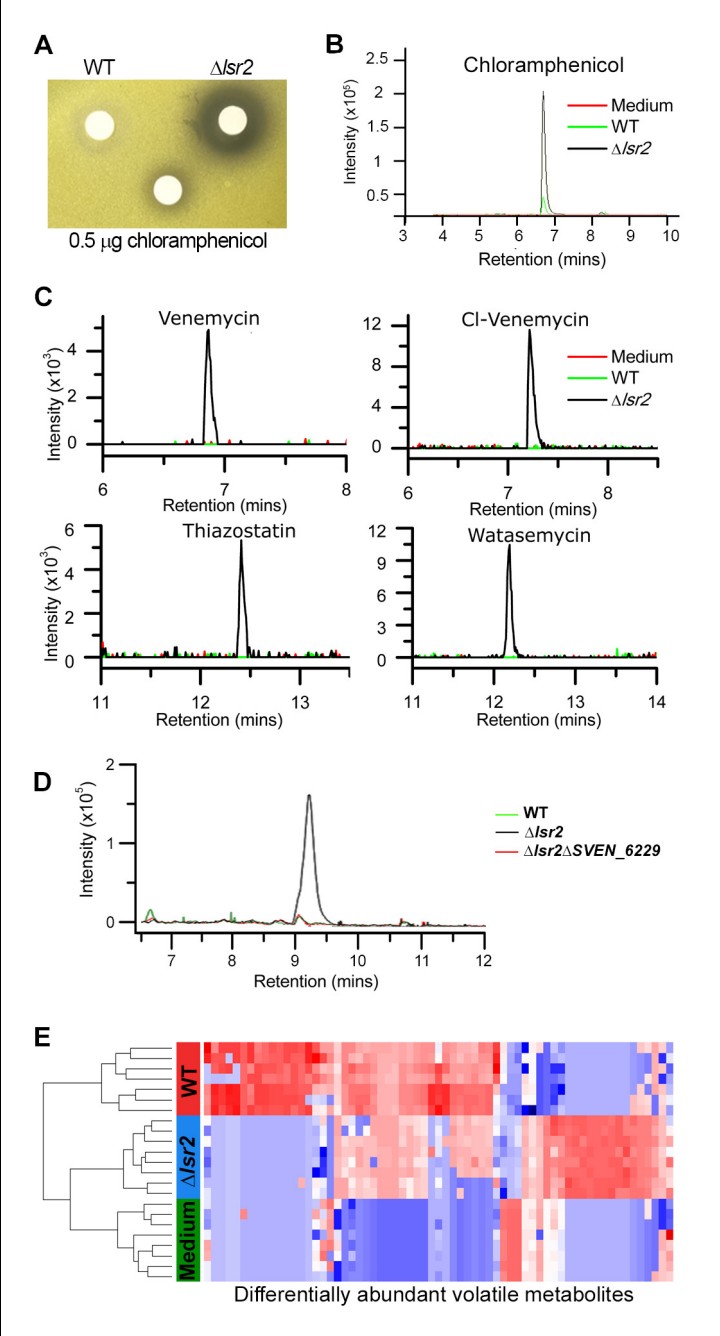

**Figure 5.** Specialized and volatile metabolite comparisons between wild type and *lsr2* mutant strains. (**A**) Bioactivity of *S. venezuelae* extracts against *Micrococcus luteus*. Wild type and *lsr2* mutant strains were cultured for 18 hr prior to extraction in methanol and reconstitution in DMSO. Extracts were applied to Whatman filter discs, alongside a chloramphenicol positive control. (**B**) Extracted ion chromatogram for chloramphenicol (*m/z* 321.005), from LC/MS analysis of methanol extracts from wild type and *lsr2* mutant cultures grown in MYM liquid medium for 3 d, alongside a medium (MYM) control. (**C**) Extracted ion chromatograms of [M-H]$^-$=219.040 (venemycin); [M-H]$^-$=252.992 (chlorinated venemycin); [M + H]$^+$ = 353.099 (watasemycin); and [M + H]$^+$ = 339.083 (thiazostatin), from LC/MS analyses of methanol extracts of wild type and *lsr2* mutant strains grown for 3 d in MYM liquid medium. (**D**) Extracted ion chromatogram of *m/z* 281.068, from LC/MS analysis of methanol extracts of wild type, *lsr2* mutant and double *lsr2 sven_6229* mutant strains, grown in MYM liquid medium for 3 d. (**E**) Heat map depicting the 65 volatile compounds (columns) that were significantly different in relative abundance (p<0.05 after BH correction) between wild type (WT, red) and *lsr2* mutant (Δ*lsr2*, blue) strains. Sterile media (Media, green) is included for comparison. Cell color corresponds to relative compound abundance after log$_{10}$-transformation,

*Figure 5 continued*
mean-centering, and unit-scaling, ranging from low abundance (blue) to high abundance (red). Dendrogram (left) was constructed using Euclidean distance as the distance metric.
DOI: https://doi.org/10.7554/eLife.47691.014
The following figure supplements are available for figure 5:
**Figure supplement 1.** Differential metabolic phenotype heat map for wild type, *lsr2, lsrL* and *lsr2/lsrL* mutant strains.
DOI: https://doi.org/10.7554/eLife.47691.015
**Figure supplement 2.** Comparison of unknown *m/z* 281 species to synthetic *N*-acetyl-5-chloro-L-tryptophan and *N*-acetyl-7-chloro-L-tryptophan.
DOI: https://doi.org/10.7554/eLife.47691.016
**Figure supplement 3.** Deletion of a halogenase-encoding gene leads to loss of multiple molecules.
DOI: https://doi.org/10.7554/eLife.47691.017

## Lsr2 alters the volatile metabolome of *S. venezuelae*

As *S. venezuelae* also produces volatile compounds with important biological roles (*Jones et al., 2017*), we compared the volatile molecules produced by wild type and *lsr2* mutant strains. After eliminating peaks associated with the growth medium, 742 discrete peaks were detected for both strains. Of these, 65 were reproducibly differentially expressed, with 38 being more abundant in the wild type, and 27 more abundant in the *lsr2* mutant (*Figure 5E*; *Supplementary file 6*), suggesting that volatile metabolites may not be subject to the same regulatory controls as other specialized metabolites. Generally, those compounds present at higher levels in the wild type had terpene-like properties. Notably, a terpene-encoding cluster (*sven_7101–7117*) was amongst a handful of metabolic clusters whose expression decreased in the absence of Lsr2 (*Table 1*; *Supplementary file 1*). In contrast, the volatile metabolites that were more abundant in the *lsr2* mutant appeared to be enriched for by-products of specialized metabolic precursors (*e.g.* derivatives of pyruvate and acetyl-CoA).

## Modulating Lsr2 activity stimulates new metabolite production in diverse *Streptomyces* species

The dramatic increase in metabolic production by the *lsr2* mutant in *S. venezuelae* prompted us to test whether it was possible to exploit this activity and stimulate new metabolite production in other streptomycetes. In *M. tuberculosis*, a dominant negative allele of *lsr2* has been reported, in which a conserved Arg residue in the C-terminal DNA-binding domain is changed to an Ala residue (*Gordon et al., 2008*). We constructed an equivalent *Streptomyces* variant (R82A mutant). Using EMSAs, we confirmed that this protein was defective in its ability to bind DNA, and that it interfered with DNA binding by the wild type protein (*Figure 6—figure supplement 1*). We also cloned this dominant negative allele behind a highly active, constitutive (*ermE\**) promoter on an integrating plasmid vector (*Figure 6A*), and introduced this 'Lsr2 knockdown' construct into wild type *S. venezuelae* to test whether it was able to phenocopy the *lsr2* mutant. Using a bioassay, we detected increased antibiotic production for this strain, relative to one carrying an empty plasmid vector (*Figure 6—figure supplement 1*). We also introduced the construct into the well-studied *S. coelicolor* strain, and observed copious production of the blue pigmented metabolite actinorhodin when grown on a medium where this compound is not typically produced (*Figure 6B*). Finally, we tested the construct in a small library of wild *Streptomyces* isolates. We screened for new metabolite production using a bioassay to assess antibiotic production. We first introduced the Lsr2 knockdown construct into strain WAC4718. This led to a significant increase in growth inhibition of *M. luteus,* and new growth inhibition of *B. subtilis,* relative to the plasmid-alone control strain (*Figure 6C*). We next introduced the knockdown and control constructs into four additional wild isolates (*Figure 6D*), and tested their antibiotic production capabilities against the indicator strain *M. luteus.* We observed new and/or increased antibiotic production for two strains (WAC7072 and WAC7520), no change in growth inhibition for one strain (WAC5514), and reduced activity in the final strain (WAC6377). Notably, these strains did not grow appreciably differently compared with their empty plasmid-containing parent strain (*e.g.* *Figure 6—figure supplement 2*). These results suggested our construct had the

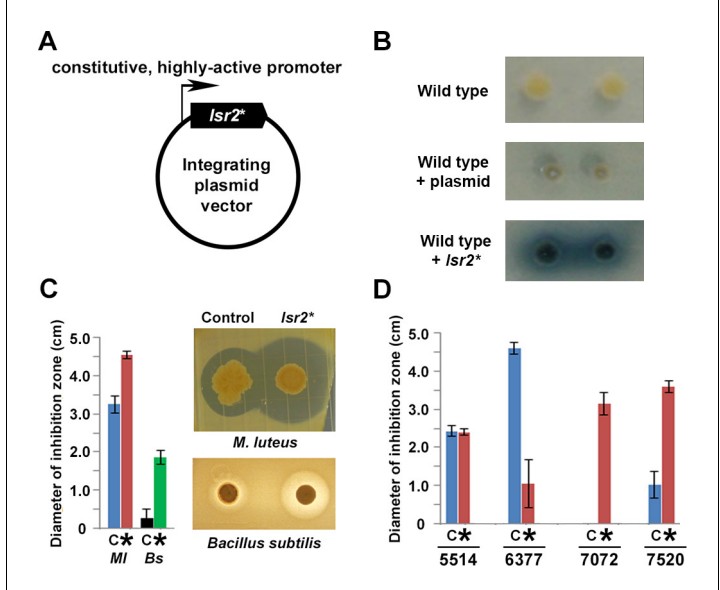

**Figure 6.** Manipulating Lsr2 activity can stimulate new antibiotic production in diverse *Streptomyces*. (A) Lsr2 activity 'knockdown' construct, where a DNA-binding defective variant of Lsr2 (*lsr2\**) is under the control of a constitutive (*ermE\**) promoter, and is on a plasmid vector bearing an apramycin resistance marker, that is capable of integrating into the chromosomes of most, if not all, streptomycetes. (B) Growth of single *Streptomyces coelicolor* colonies on Difco nutrient agar. Top: wild type; middle: wild type carrying the empty plasmid; bottom: wild type carrying the plasmid with dominant negative *lsr2\** variant. (C) Antibiotic bioassay using the wild *Streptomyces* strain WAC4718, bearing either the control (empty) plasmid (C), or the Lsr2 knockdown construct (\*), tested against the indicator strains *M. luteus* (*Ml*) or *Bacillus subtilis* (*Bs*). Bars indicate standard error (*n* = 4). (D) Antibiotic bioassay using four different wild *Streptomyces* strains (WAC5514, WAC6377, WAC7072, and WAC7520), carrying either empty plasmid (C) or the Lsr2 knockdown construct (\*), tested against *M. luteus* as an indicator strain. Bars indicate standard error (*n* = 4).

DOI: https://doi.org/10.7554/eLife.47691.018

The following figure supplements are available for figure 6:

**Figure supplement 1.** Dominant negative Lsr2 variant (Lsr2\*) inhibits DNA binding by wild type Lsr2 and promotes antibiotic production in *S. venezuelae*.

DOI: https://doi.org/10.7554/eLife.47691.019

**Figure supplement 2.** Growth curves of WAC04718 containing either an empty plasmid or one expressing Lsr2\*.

DOI: https://doi.org/10.7554/eLife.47691.020

ability to downregulate Lsr2 activity in a wide range of streptomycetes, and could serve as a broadly applicable means of stimulating antibiotic production in these bacteria.

## Discussion

The nucleoid-associated protein Lsr2 has been tied to virulence and environmental adaptation in *Mycobacterium*, and like H-NS in *E. coli,* it has been proposed to function to repress the expression of 'foreign' DNA (*Gordon et al., 2010*; *Gordon et al., 2011*). Here, we demonstrate a role for Lsr2 in repressing the expression of laterally acquired sequences in *Streptomyces*, as well as in suppressing the expression of antisense RNA, as has also been observed for H-NS. Uniquely in *Streptomyces,* however, it appears that Lsr2 function has been co-opted for the control of specialized metabolism, and that the cryptic/silent nature of many of these metabolic clusters is due to direct Lsr2 repression.

### Mechanism of Lsr2-mediated repression

Previous work on Lsr2 from the mycobacteria has shown Lsr2 preferentially binds AT-rich DNA (*Gordon et al., 2010*; *Gordon et al., 2011*), and our findings suggest that this is also the case in the

streptomycetes. Unlike more conventional transcription factors, we found that Lsr2 binding sites in *S. venezuelae* tended to be quite broad, centring on AT-rich sequences, extending hundreds (or thousands) of base-pairs, and frequently encompassing promoter regions (*Supplementary file 3*). In the proteobacteria, H-NS can polymerize along the chromosome (*Liu et al., 2010*; *Lim et al., 2012*; *Will et al., 2018*), repressing transcription by shielding the DNA from binding by transcription factors or RNA polymerase. It can also bridge disparate DNA segments (*van der Valk et al., 2017*), repressing gene expression by trapping RNA polymerase, and/or changing the local DNA architecture. Lsr2 repression in *S. venezuelae* is consistent with both polymerization and bridging mechanisms. The larger binding sites, often associated with transcriptional changes, could be the result of Lsr2 filamentation along the chromosome in those regions. We also identified multiple specialized metabolic clusters having more than one Lsr2 binding site (see *Figure 3*). This was particularly notable within the right arm of the chromosome (*Table 1*). These sites were often smaller (*Supplementary file 3*), and it is possible that gene repression is achieved through bridging between these sites.

Many of the Lsr2 binding sites identified here, however, were not associated with altered transcription of their flanking genes. It is conceivable that these sites serve more of an architectural role, with Lsr2 binding promoting chromosome organization and compaction. Binding at these sites may also exert indirect effects on transcription, as a result of altered DNA structure and accessibility.

In this study, we focussed our attention on the DNA-binding activity of Lsr2, but it is worth noting that post-transcriptional regulatory roles have been identified for related proteins. In particular, H-NS can also bind RNA (*Park et al., 2010*), where it stabilizes target transcripts and promotes their translation. Notably, within the differentially expressed genes identified here, Lsr2 enhanced the expression of 10% of these genes. Their activation by Lsr2 (whether it be by binding their transcripts, or some other means) makes them interesting candidates for future investigations aimed at understanding alternative regulatory roles for Lsr2.

## Role for Lsr2 repression

Our results suggest that Lsr2 functions both as a 'genome sentinel', and as a central governor of specialized metabolism. It represses the expression of many genes that seem to have been recently acquired based on conservation analysis. However, it also controls the expression of many well-conserved clusters in *S. venezuelae*. It is not clear how *Streptomyces* species acquire their specialized metabolic clusters. The clustered nature of specialized metabolic genes makes them amenable to transfer between species through conjugation or transduction (*Streptomyces* are not naturally competent for DNA transformation), although genomic studies have suggested that the transfer and maintenance of entire clusters is relatively uncommon (*McDonald and Currie, 2017*). In many clusters, the pathway-specific regulators (*Williamson et al., 2006*; *Fernández-Martínez et al., 2014*) and/or resistance determinants (e.g. *Flatt and Mahmud, 2007*; *Thaker et al., 2013*; *Mak et al., 2014*) are encoded near the ends of the cluster. Loss of either of these genetic elements during cluster transfer to a recipient could lead to inappropriate cluster expression in the absence of a fail-safe mechanism provided by proteins like Lsr2.

Within *S. venezuelae*, seven clusters lack an obvious pathway-specific regulator, and of these, five are under Lsr2 control. Similarly, six of nine clusters lacking an associated transporter are affected by Lsr2 (although resistance can be conferred by means other than transport, and not all specialized metabolites function extracellularly). In the streptomycetes, Lsr2 may therefore act to protect the cell from the toxic products of newly acquired clusters, until they are either integrated into existing regulatory networks or are lost from the cell. Widely conserved clusters controlled by Lsr2 (e.g. siderophore/desferrioxamine and bacteriocin) likely represent instances of successful integration, where Lsr2 control has evolved such that its repression can be alleviated under appropriate conditions. However, not all specialized metabolic clusters are under Lsr2 control. The products of Lsr2-independent clusters may be important for growth under laboratory conditions (and thus any Lsr2-mediated repression may have been alleviated under our experimental conditions), or they may be largely benign and/or have a low fitness cost associated with their production. On the other end of the spectrum are the volatile compounds, many of which required Lsr2 for their production. It appears that the synthesis of these molecules may be subject to different regulatory constraints than those of the specialized metabolites.

## Control of Lsr2 expression and activity

As Lsr2 governs the expression of the majority of specialized metabolic clusters in *S. venezuelae,* it is critical to understand how its expression and activity are controlled. In the proteobacteria, factors that impact H-NS expression and activity have been extensively studied; however, far less is known about what affects Lsr2 levels and function in the actinobacteria.

In the proteobacteria, *hns* expression is activated by multiple transcription factors (*Falconi et al., 1996*; *La Teana et al., 1991*), and is negatively regulated by both small RNAs and H-NS itself (*Falconi et al., 1993*; *Lalaouna et al., 2015*). In the actinobacteria, there is currently nothing known about the transcriptional regulation of *lsr2*, although it likely governs its own expression: there is a large Lsr2 binding site that overlaps the *lsr2* promoter (*Supplementary file 3*), suggesting that, like H-NS, it negatively regulates its own expression. How *lsr2* expression is activated, and whether it is also subject to post-transcriptional regulation remains to be seen.

At a protein level, H-NS activity can be modulated by interaction with a multitude of proteins, including association with paralogous proteins like StpA (*Müller et al., 2010*). Intriguingly, all streptomycetes encode a paralogous Lsr2-like protein, termed LsrL. Our data suggest that there exists regulatory interplay between these proteins, with Lsr2 repressing *lsrL* expression. It will be interesting to see whether LsrL associates with Lsr2 to form hetero-oligomers, and whether such an association alters Lsr2 activity. Deleting *lsrL* did not have profound phenotypic consequences, at least under the conditions we tested, so understanding its biological role in *Streptomyces* will require additional investigation. Unlike the streptomycetes, the mycobacteria do not encode additional Lsr2-like proteins. However, recent work in *M. tuberculosis* has suggested that Lsr2 can associate with the unrelated nucleoid-associated protein HU (*Datta et al., 2019*); whether an equivalent interaction occurs in *Streptomyces* has yet to be determined. Lsr2 also appears to be subject to post-translational modification, having been identified in several phospho-proteome screens conducted in *Streptomyces coelicolor* (*Parker et al., 2010*; *Manteca et al., 2011*), although how phosphorylation affects Lsr2 activity is currently unclear.

H-NS-mediated repression can be alleviated through competition for binding to similar sequences by other transcription factors (*Will et al., 2015*). Equivalent 'counter-silencing' mechanisms have been reported for Lsr2 in the mycobacteria (*Kurthkoti et al., 2015*). Given that at least a subset of the Lsr2-controlled specialized metabolic clusters are expressed under particular growth conditions in *S. venezuelae* (*e.g.* desferrioxamine, melanin), there must exist mechanisms to relieve Lsr2 repression in the streptomycetes as well. Intriguing candidates for this could include global antibiotic regulators, or cluster-situated regulators, and revealing how Lsr2 is integrated into the larger regulatory networks governing specialized metabolism will be a major priority.

## Chromosome organization and the regulation of specialized metabolism

Lsr2 is one of multiple nucleoid-associated proteins encoded by the streptomycetes, including sIHF (*Swiercz et al., 2013*), the HU proteins HupA and HupS (*Salerno et al., 2009*), and BldC (*Bush et al., 2019*). To date, only BldC binding and regulation has been thoroughly characterized, but unlike Lsr2, its primary regulatory targets are developmental determinants, not specialized metabolic clusters. Phenotypic analyses of *sIHF* and *hupS* mutants in *S. coelicolor* revealed major sporulation defects, in contrast to the modest developmental delay observed for the *lsr2* mutant (*Salerno et al., 2009*; *Swiercz et al., 2013*). Both mutations also impacted metabolism, with the *sIHF* mutant exhibiting both enhanced and reduced production of pigmented antibiotics, depending on media conditions, and the *hupS* mutant failing to produce the brown spore pigment in *S. coelicolor*. While a comprehensive analysis remains to be conducted, neither sIHF nor HupS appear to function like Lsr2 in exerting global control over specialized metabolism.

In the fungi, chromosome organization is governed by the histones which, like the nucleoid-associated proteins in bacteria, function to both compact the chromosome and control gene expression. Like *Streptomyces,* fungi possess a multitude of cryptic secondary metabolic clusters, and many of these are silenced as a result of histone activity (*Keller, 2019*). Successful cluster activation has been achieved by manipulating histone activity through altered acetylation or methylation (*Pfannenstiel and Keller, 2019*).

Given the broad conservation of Lsr2 across the streptomycetes, and its significant impact on specialized metabolism in these bacteria, Lsr2 is an attractive candidate for activity modulation, like the histones in fungi. Deleting *lsr2* may not be feasible in all *Streptomyces,* given recent studies suggesting that it is an essential gene in some species (*Najah et al., 2019*). However, our work here suggests that downregulating Lsr2 activity may offer an effective approach to alleviating metabolic repression, and can profoundly alter the metabolic output of a wide range of streptomycetes. Collectively, it may provide a new avenue for accessing otherwise cryptic natural products in these metabolically gifted bacteria, facilitating both our understanding of the chemical ecology associated with microbial signalling and interactions, and our ability to identify new compounds for clinical development.

## Materials and methods

### Bacterial strains, plasmids, oligonucleotides and culture conditions

All strains and plasmids/cosmids are outlined in *Supplementary file 7*, while oligonucleotide information is provided in *Supplementary file 8*. *S. venezuelae* strains were grown at 30°C on MS (soy flour-mannitol) agar, MYM (maltose-yeast extract-mannitol) agar, YPD (yeast-peptone-dextrose) agar, and YP (yeast-peptone) agar, or in liquid MYM medium prepared as described previously (*Kieser et al., 2000*; *Jones et al., 2017*). *S. coelicolor* strains were grown on Difco nutrient agar plates, while wild *Streptomyces* isolates were grown on ISP4 medium (Difco) supplemented with maltose (1 g/L), mannitol (1 g/L), sucrose (1 g/L) and glycerol (1 g/L), or in Bennet's medium. *E. coli* strains were grown at 37°C on or in LB (Luria-Bertani) medium or in liquid SOB (Super Optimal Broth) (*Kieser et al., 2000*).

### Strain and plasmid construction

An in-frame deletion of *lsr2* (*sven_3225*) was created using the ReDirect PCR targeting method (*Gust et al., 2003*). The *lsr2* coding region was replaced with the *aac(3)IV-oriT* resistance cassette, which was subsequently excised using the yeast FLP recombinase to leave an 81 bp scar. The *aac(3) IV-oriT* cassette was amplified from pIJ773 using the primer pair Sven3225disruptF and Sven3225disruptR2 to generate an extended resistance cassette (oligonucleotide sequences listed in *Supplementary file 8*). Cosmid 1-C1 (http://strepdb.streptomyces.org.uk/) was introduced into *E. coli* BW25113 containing pIJ790, and the *lsr2* coding region was replaced with the extended resistance cassette. Cosmid 1-C1Δ*lsr2::aac(3)IV-oriT* was confirmed both via PCR using the flanking primers sven3225F2 and sven3225R2, and through a diagnostic restriction digest. The modified cosmid was then introduced into *S. venezuelae* by conjugation. Two representative apramycin-resistant, kanamycin-sensitive null mutants were selected for morphological analysis. Cosmid 1-C1Δ*lsr2::aac(3) IV-oriT* was introduced into *E. coli* BT340 in which the FLP recombinase was induced to excise the *aac(3)IV-oriT* cassette from the cosmid. The cosmid backbone was then targeted to replace *bla* with the *hyg-oriT* cassette from pIJ10701 (*Gust et al., 2004*). The resulting cosmid was checked using PCR (*Supplementary file 8*) and restriction digest, prior to being mobilised into *S. venezuelae* Δ*lsr2::aac(3)IV-oriT.* Hygromycin-resistant exconjugants were selected, and then screened for a double cross-over event resulting in aparamycin-sensitive, hygromycin-sensitive scarred mutants that were confirmed by PCR (*Supplementary file 8*).

*S. venezuelae* Δ*lsr2* was complemented through cloning *lsr2,* together with 257 bp upstream and 293 bp downstream flanking sequences, into the *Eco*RV-digested integrative plasmid pIJ82. Gene synthesis (GenScript) was used to generate a C-terminal triple FLAG-tagged variant (DYKDHDGD YKDHDIDYKDDDDK, separated from the Lsr2 sequence by a triple glycine linker) bearing the same upstream and downstream sequences as the native complementation construct. This synthesized sequence was flanked by *Bgl*II sites, which facilitated subcloning into the *Bam*HI site of the integrative plasmid pIJ10706 (pIJ82 and pIJ10706 are identical except that pIJ10706 uses the *aac(3)IV* promoter to drive expression of *hyg*). Both *lsr2*-carrying plasmids were introduced individually into the *lsr2* mutant and assessed for their ability to complement the developmental delay observed on solid MYM.

An in-frame deletion of *lsrL* (*sven_3832*) was created using the ReDirect PCR targeting method (*Gust et al., 2003*) described above. The *lsrL* coding region was replaced with the *aac(3)IV-oriT*

resistance cassette in cosmid 4E19 and conjugated from the non-methylating *E. coli* strain ET12567 (*MacNeil et al., 1992*) containing pUZ8002 (*Paget et al., 1999*) into *S. venezuelae*. Two representative apramycin-resistant, kanamycin-sensitive null mutants were confirmed by PCR (*Supplementary file 8*) and were subjected to morphological and metabolic analyses.

To mutate *sven_6229*, CRISPR-Cas-mediated mutagenesis was used (*Cobb et al., 2015*), with minor alterations to the published protocol. Briefly, a 32 nucleotide deletion, along with an in-frame stop codon, was introduced into *sven_6229*. The guide RNA sequence was cloned into the *Bsb*I site of pCRISPomyces2, following the annealing of the overlapping oligonucleotides Sven6229 GuideF and Sven6229 GuideR (*Supplementary file 8*). The editing template was generated by first amplifying fragments upstream (Sven6229 UpF/R) and downstream (Sven6229 DownF/R) of the guide RNA sequences. These sequences were then joined by overlap extension PCR, before being digested and cloned into the *Xba*I site of the guide RNA-carrying pCRISPomyces vector. Sequence integrity of both the guide RNA and editing template was confirmed by sequencing. The resulting plasmid was conjugated into the *lsr2* mutant (*Supplementary file 7*), and exconjugants were selected for using apramycin and nalidixic acid. Colonies were then streaked on MS agar plates without antibiotic supplementation, and were screened for the desired deletion using the Sven6229 GuideF and Sven6229 DownR primers. Candidate deletion mutants were subjected to a final PCR check, using Sven6229 UpR and Sven6229 ConR, and the resulting product was sequenced to confirm the mutation.

To investigate the effects of AT-content on Lsr2 binding and gene expression, we focussed on a validated Lsr2 binding site between *sven_5106* and *sven_5107*, where the expression of these genes was increased upon loss of Lsr2, suggesting Lsr2 repression. To clone a ~9 kb DNA fragment spanning *sven_5105–5107*, the TOPO TA cloning kit was used as per the manufacture's instructions. Briefly, the fragment was amplified using the Phusion proofreading polymerase (New England Biolabs) with oligonucleotides Sven5105_5107F and Sven5105_5107R (*Supplementary file 8*), a 72°C annealing temperature, and cosmid Sv-3-D04 (*Supplementary file 7*) as template. The amplified product was purified by gel extraction, and was then incubated with *Taq* polymerase and dATP at 72°C for 15 min. Four microlitres of the resulting A-tailed product were mixed with salt solution and pCR2.1-TOPO vector provided in the cloning kit, before being introduced into Subcloning Efficiency DH5α competent cells (ThermoFisher Scientific). The *sven_5105–5107* containing plasmid was verified using restriction enzyme digestion and sequencing. To create mutant variants, synthetic gene fragments were generated and amplified using oligonucleotides Sven5106_5107F and Sven5106_5107R (*Supplementary file 8*). The amplified products were cloned between unique *Nhe*I and *Avr*II sites within the *sven5105-07* sequences. The designed mutations were confirmed by restriction digestion and sequencing. All validated *sven_5105–5107* variants (wild type and mutants) were excised from the TOPO vector using *Xba*I and *Spe*I, and cloned into the *Spe*I site of pRT801. Constructs were then conjugated into wild type *S. venezuelae* and Δ*lsr2* mutant strains (for expression analysis), and the Δ*lsr2* mutant strains complemented with either *lsr2* or *lsr2−3× FLAG* (for ChIP analyses).

### *Streptomyces* cell extract preparation, SDS-PAGE, and immunoblotting

Cell extracts were prepared from a 1 mL aliquot of *S. venezuelae* cells grown in liquid MYM medium. The protein extracts were separated using 15% SDS-PAGE and were stained with Coomassie brilliant blue R-250. Equivalent amounts of total protein were loaded onto a second 15% SDS-PAG, and following transfer to PVDF membranes, were subjected to immunoblotting with anti-FLAG antibodies (1:1,500; Sigma) and anti-rabbit IgG horseradish peroxidase (HRP)-conjugated secondary antibodies (1:3,000; Cell Signaling).

### Lsr2 overexpression, purification and electrophoretic mobility shift assays (EMSAs)

*lsr2* amplified using primers NdeISven3225F and BamHISven3225R (*Supplementary file 8*) was digested and the product cloned into the similarly digested pET15b (*Supplementary file 8*). After sequencing, this construct was introduced into *E. coli* Rosetta cells (*Supplementary file 8*). Overexpression of 6× His-*lsr2* was achieved by growing cultures at 37 °C to an $OD_{600}$ of 0.5, and then adding 0.5 mM IPTG (isopropyl β- D-1-thiogalactopyranoside). Cells were grown for a further 3 hr at 30 °C before harvesting and resuspending in binding buffer (50 mM $NaH_2PO_4$, 300 mM NaCl and 10

mM imidazole, pH 8.0) containing 1 mg/mL lysozyme and one complete mini EDTA-free protease inhibitor pellet (Roche). Cell suspensions were incubated on ice for 20 min before sonication. $6\times$ His-Lsr2 was purified by binding to 1 mL Ni-NTA agarose (Invitrogen), after which the resin was collected and the bound protein was washed with binding buffer supplemented with increasing concentrations of imidazole. Purified proteins were ultimately eluted using 500 mM imidazole. Purified protein was exchanged into storage buffer (20 mM Tris-HCl, pH 8, 150 mM NaCl, 25% glycerol and 1.4 mM β-mercaptoethanol) using an Amicon Ultra-15 Centrifugal Filter with a 3 kDa cut-off. Bradford assays were conducted to measure protein concentrations.

EMSAs were performed using 124–280 bp probes amplified by PCR and 5'-end-labeled with [γ-$^{32}$P]dATP (primers prefixed 'emsa' are listed in *Supplementary file 8*). Increasing concentrations of Lsr2 (0–5 µM) were combined with either 1 or 10 nM probe, 1 mg/mL bovine serum albumin (BSA) and binding buffer (10 mM Tris, pH 7.8, 5 mM MgCl$_2$, 60 mM KCl and 10% glycerol). Each reaction was incubated for 10 min at room temperature, followed by 30 min on ice prior to adding a glycerol-based loading dye and running on a 12% native polyacrylamide gel. To test binding specificity, competition assays were established in which increasing concentrations (0–160 nM) of unlabeled probe were added together with 4 nM labeled probe and 1 µM Lsr2, to the EMSA reactions described above. Gels were exposed to a phosphor plate for 1 hr, before being visualized using a phosphorimager (Typhoon FLA 9500).

## RNA isolation and RT-(q)PCR

Wild type *S. venezuelae* and the Δ*lsr2* mutant strain were grown in 300 mL MYM cultures in duplicate. After 8 hr (vegetative growth), 12 hr (early mycelial fragmentation) and 18 hr (late mycelial fragmentation/sporulation), density at OD$_{450}$ was measured, and a 60–90 mL sample was harvested. Subsequent experiments involved growing wild type and Δ*lsr2* mutant strains carrying *sven_5105-sven_5107* variants on an integrating plasmid. These strains were grown in duplicate, in 50 mL MYM liquid medium for 18 hr. In all cases, RNA was isolated as described in Moody et al. (*Moody et al., 2013*), using a modified guanidium thiocyanate protocol (*Chomczynski and Sacchi, 1987*). Primers HrdBF and HrdBR, or SVEN4987F/SVEN4987R (*Supplementary file 8*) were used for PCR checks, alongside a quantified chromosomal DNA control, to confirm any DNA contamination was <0.005%.

Reverse transcription (RT) reactions were performed as described previously (*Haiser et al., 2009*; *Moody et al., 2013*). In brief, gene-specific reverse primers (*Supplementary file 7*), or random oligonucleotides were annealed to 1 µg of total RNA prior to cDNA synthesis using SuperScript III reverse transcriptase (Invitrogen) (wild type and mutant) or Lunascript RT (New England Biolabs) (*sven5105-5107* variants), respectively.

To validate RNA-sequencing results, two microlitres of the resulting cDNA were used as template DNA for PCR, with a 58°C annealing temperature. The number of cycles was optimized to ensure that amplification was occurring within the linear range of the reaction (33 cycles for *sven_0514*, *sven_6216*, *sven_6264* and *hrdB*, and 30 cycles for *sven_0493* and *sven_5135*). Negative control reactions were run to confirm the absence of genomic DNA contamination in the RNA samples, and involved adding an equivalent volume of a reverse transcription reaction in which nuclease free water had been added in place of reverse transcriptase. All reverse transcription reactions and PCR amplifications were carried out in duplicate, using RNA isolated from two independent cultures.

For the *sven_5105–5107* variant-containing strains, 2.5 µL of cDNA (1:4) were used as template for qPCR. Primers 5106F/5106R were used to amplify target gene from cDNA with a 55°C annealing temperature. 'No RT' samples were run to confirm no DNA contamination. All samples were assessed in biological duplicate and technical triplicate. qPCR data were normalized to *rpoB* and were analyzed using a relative quantification method ($2^{-DDC_T}$) (*Livak and Schmittgen, 2001*).

## RNA-seq sample preparation and data analysis

Library construction and sequencing were performed by the Farncombe Metagenomics Facility, McMaster University, Hamilton, Canada. Total RNA (1.7 µg) from each sample was subjected to rRNA depletion using RiboZero (Epicentre), as per the manufacturer's instructions. Library preparation was performed using the NEBNext Ultra Directional RNA Library Prep Kit (NEB), including double-AMPure bead (Beckman Coulter) size selection. Following quality control, libraries were pooled

in equimolar amounts and sequenced over two lanes of the HiSeq 1500 using the TruSeq Rapid (v1) chemistry with onboard cluster generation and a 1 × 75 bp protocol.

Raw sequencing reads were trimmed to remove low-quality 3' ends using PrinSeq (*Schmieder and Edwards, 2011*). Trimmed reads were checked for quality using FastQC (www.bio-informatics.babraham.ac.uk/projects/fastqc/) and aligned to the *S. venezuelae* ATCC 10712 genome sequence using Bowtie2 (*Langmead et al., 2009*).

The resultant SAM files were converted to BAM format, sorted by genomic position and indexed to create BAI files (*Li et al., 2009*). The BAM files were analyzed both visually using Integrated Genomics Viewer (Version 2.3.60) (*Robinson et al., 2011*), and using Rockhopper2 (*Tjaden, 2015*). We assigned a cut-off for significance using a *p*-value adjusted for multiple testing that was less than 0.01 (*q*-value), and filtered for genes displaying a fold change greater than four.

## Chromatin immunoprecipitation

*S. venezuelae* Δ*lsr2* was complemented using an integrating plasmid (pIJ10706/pIJ82) carrying either wild type *lsr2* or *lsr2–3 × FLAG* (*Supplementary file 7*). Each culture was then grown in 300 mL MYM cultures in duplicate. After 18 hr, the density at OD$_{450}$ was measured and the developmental progression of each strain was monitored by light microscopy. A 1 mL sample was then taken for immunoblot analysis, and an 80 mL sample was transferred to a sterile flask. Formaldehyde was added to a final concentration of 1% (vol/vol) to cross-link protein to DNA, after which cultures were incubated for a further 30 min. Glycine was then added to a final concentration of 125 mM. Immuno-precipitation was carried out as described in Bush *et al.* (*Bush et al., 2013*), using Anti-FLAG (DYKDDDDK) affinity gel (BioTools). Immunoprecipitation, and subsequent sequencing, were done in duplicate.

Library construction and sequencing were performed by the Farncombe Metagenomics Facility, McMaster University, Hamilton, Canada. The NEBNext Ultra DNA Library Prep Kit was used for library preparation, starting with 10 ng of the sheared ChIP DNA and including a double-AMPure bead (Beckman Coulter) size selection. Following quality control, libraries were pooled in equimolar amounts and sequenced on one MiSeq run using a 2 × 75 bp (v3) configuration.

## ChIP-seq data analysis

The reads in the fastq files were first checked for quality using FastQC (www.bioinformatics.babra-ham.ac.uk/projects/fastqc/), then aligned to the *S. venezuelae* ATCC 10712 genome (GenBank accession number NC_018750) using Bowtie2 (*Langmead et al., 2009*). The resultant SAM files were converted to BAM format, sorted by genomic position and indexed to create BAI files (*Li et al., 2009*). The BAM files were visualized using Integrated Genomics Viewer (Version 2.3.60) (*Robinson et al., 2011*), and were subjected to quantitative analysis.

*MACS2* was run from the command line to normalize all *lsr2* and the *lsr2–3 × FLAG* samples against total DNA with the mappable genome size set at 7.92 × 10$^6$ (90% of the *S. venezuelae* genome) to generate BED files (*Zhang et al., 2008*). The BED files were in turn used to generate a CSV sample sheet that was read into the R package for statistical computing (*R Development Core Team, 2013*) using the *read* function of the *DiffBind* package (*Stark and Brown, 2011*). The *dba.count* function of the *DiffBind* package was used to calculate a binding matrix with scores based on read counts for each sample. The *dba.contrast* function of the *DiffBind* package was then used to compare the *lsr2* (negative control) samples with the *lsr2–3 × FLAG* samples. The *dba.analyze* function of the *DiffBind* package was used to run an *edgeR* analysis that identified sites that were significantly differentially bound [having a *p*-value adjusted for multiple testing that was less than 0.01 (*q*-value)]. These differentially bound sites were retrieved using the *dba.report* function.

## ChIP-quantitative PCR

Strains were grown in 10 mL of MYM medium overnight, before being subcultured in duplicate into 50 mL of MYM medium. After incubating for 18 hr, formaldehyde was added to a final concentration of 1% (v/v) to cross-link protein to DNA. The cultures were incubated for an additional 30 min, at which time glycine was added to a final concentration of 125 mM. Immunoprecipitation was performed as described above, only using the FLAG M2 antibody (Sigma).

To quantify the abundance of target genes of interest in the ChIP DNA, 20 µL qPCR reactions were prepared using Luna Universal qPCR Master Mix (New England Biolabs) and 2.5 µL of ChIP DNA (1:10) as template. Primers 0926F/0926R and 5105F/5105R (*Supplementary file 8*) were used to amplify target genes from ChIP DNA with a 55°C annealing temperature. qPCR data were then analyzed using DART-PCR (*Peirson, 2003*).

## Phylogenetic analysis of *Streptomyces* species

Phylogenetic analysis was conducted based on the concatenated protein sequences encoded by selected single-copy phylogenetic marker genes *serS, rpoB, secY,* and *rplB*. These sequences were extracted from the complete genome sequences of the different streptomycetes, as well as from an outgroup (*Bacillus subtilis*), all of which were accessed using the NCBI database. Alignments were generated using ClustalX2 (*Larkin et al., 2007*) using a neighbor joining cluster algorithm, with iteration at each alignment step and 1000 bootstrap replications. The phylogenetic tree was visualized using FigTree v1.4 (http://tree.bio.ed.ac.uk/software/figtree).

## Specialized metabolite extraction and analysis

*S. venezuelae* WT and Δ*lsr2* strains were grown in triplicate as a lawn on MYM agar over a time-course of 3, 5 and 7 days. The entirety of each plate (as well as an MYM agar control) was macerated in 25 mL *n*-butanol, sonicated for 5 min in a Branson 2520 tabletop ultrasonic cleaner and shaken overnight at 4°C. The agar was removed by passing through a milk filter, then Whatman filter paper, after which the solvent was split into two aliquots and dried down in a GeneVac. The residue was reconstituted in either 500 µL 50% methanol for bioassays, or 1 mL of 1:1 MeCN:$H_2O$ for LC-MS analysis.

LC-MS analysis was performed on a Waters Alliance Acquity UPLC system coupled to a Xevo G2S-QTof. A 10 µL aliquot was taken and diluted with 90 µL 1:1 MeCN:$H_2O$, and 10 µL of this diluted solution were injected onto a Phenomenex Kinetex Biphenyl column (1.7 µm, 2.1 × 100 mm).

The samples were separated using a gradient of 5% to 95% acetonitrile (0.1% [vol/vol] formic acid) at 40°C over 25 min, with a flow rate of 0.4 mL/min. Positive electrospray ionization was performed and the ions were scanned over a mass range of 100 to 1200 Da. Data were analyzed using MZmine 2 software (*Pluskal et al., 2010*).

For soluble metabolites, and quantitation of the known metabolites shown in *Figure 5C*, cultures grown in MYM liquid medium for 3, 4, or 5 days were centrifuged, and the supernatant lyophilized. The lyophile was redissolved in 10 mL of 50:50 acetonitrile/water and shaken on a rotary shaker for 1 hr. After centrifugation to remove particulates, samples were analyzed on an Agilent UPLC-QTOF. Injections of 2 µL were separated on Zorbax Eclipse plus C18 column (2.1 mm x 50 mm) and a gradient of 0–1 min isocratic 100% A, 1–17 min gradient from 100–0% A, and 17–20 min isocratic 100% B, where A is 98% water, 2% methanol, 0.1% formic acid and B is 100% methanol, 0.1% formic acid, at a flow rate of 0.4 mL/min. Samples were measured in positive and negative ionization modes to gain the greatest coverage. Samples were analyzed in technical triplicates and key features were assessed in biological duplicates. Fragmentation analysis of features of interest was performed with a collision energy of 25 V.

For the soluble metabolites shown in *Figure 5D*, cultures were grown in MYM liquid medium for 3 days, after which Diaion HP-20 resin (3% w/v) was added into the cultures and shaken for 8 hr. The resin was washed using distilled water, separated from the supernatant using filter paper, and dried overnight. Metabolites were eluted from the resin using 5 mL methanol. The eluates were dried *in vacuo* and resuspended in 250 µL methanol. After centrifugation to remove particulates, the crude extracts were analyzed using an Agilent 1200 LC coupled to a Bruker micrOTOF II (ESI-MS). Injections of 5–10 µL sample were separated on a Zorbax Eclipse XDB C18 column (100 mm×2.1 mm× 3.5 µm) at a flow rate of 0.4 mL/min for 30 min. A gradient of 0–21 min from 95% to 5% solvent A, 21–26 min isocratic 5% A, and a gradient of 26–30 min from 5% to 95% A, where A is water with 0.1% formic acid and B is 100% methanol with 0.1% formic acid. At least two biological duplicates were measured in positive or negative ionization modes.

Antibiotic bioassays were performed by testing methanol extracts of *S. venezuelae* grown for 1 to 3 days against *Micrococcus luteus*. Twenty microliters of each extract was applied to a Whatman antibiotic assay disc and applied to a lawn of LB inoculated with a 25-fold dilution of the *M. luteus*

indicator strain grown to mid-exponential phase. The plates were incubated overnight at 30°C before measuring the size of the zone of clearing. Bioassays for wild *Streptomyces* were performed by growing isolates (knockdown- and plasmid-control containing) on ISP4 supplemented medium for 6 days at 30°C. Overnight cultures of indicator strains (*M. luteus* or *B. subtilis*) were mixed with 1% soft nutrient agar, which was allowed to solidify before being overlaid atop the wild *Streptomyces* strains, after which the cultures were incubated overnight at 37°C.

## Synthesis of N-acetyl-chloro-tryptophan standards

Synthesis of *N*-acetyl-5-chloro-L-tryptophan was achieved using Wang resin (50 mg, 1 mmol/g loading), which was swollen in anhydrous dimethylformamide (DMF). Fmoc-5-chloro-L-Trp-OH (57.5 mg, 0.125 mmol, 2.5 eq.) was dissolved in 5 mL 9:1 DMF:dichloromethane (DCM) and cooled to 0°C. Diisopropylcarbodiimide (DIC) (6.3 mg, 0.050 mmol, one eq.) was added in minimal DCM. The reaction was stirred for 30 min at 0°C, in a flask fitted with a drying tube. The anhydride mixture was added to the swollen resin, after which DMAP (0.6 mg, 5 μmol, 0.1 eq.) was added, and the flask was then periodically agitated at room temperature for 2 hr. The resin was washed in $3 \times 10$ mL DMF, followed by $3 \times 10$ mL DCM. The Fmoc group was removed by the addition of 10 mL 20% (V/V) piperidine in DMF, after which the suspension was agitated for 20 min. The resin was washed as above, before acetylation was carried out with the addition of acetic anhydride (5.0 μL, 0.05 mmol, one eq.) and diisopropylethylamine (DIPEA) (1 μL, 5 μmol, 0.1 eq.) in 5 mL DMF. The resulting suspension was agitated for 30 min at room temperature. The resin was washed as above, prior to cleavage being carried out with 10 mL 95% TFA, 2.5% triethylsilane, 2.5% DCM for 30 min. The eluent was then collected and evaporated to dryness. Analysis by LC-MS was completed without any further purification.

To synthesize *N*-acetyl-7-chloro-L-tryptophan, 7-chloro-L-Trp-OH (24 mg, 0.1 mmol) was dissolved in 20 μL 50 mM ammonium bicarbonate. Fifty microliters of an acetylation mixture (20 μL acetic anhydride, 60 μL methanol) were added to the amino acid, after which the mixture was agitated for 1 hr at room temperature. The solvent was evaporated to dryness and the resulting product was analyzed without additional purification.

## Analysis of volatile metabolites

Volatile metabolites in the headspace of culture supernatants were concentrated, analyzed, and relatively quantified using headspace solid-phase microextraction coupled to two-dimensional gas chromatography time-of-flight mass spectrometry (HS-SPME-GC×GC TOFMS), as described previously (*Jones et al., 2017*). Four milliliters of culture supernatants were transferred to 20 mL air-tight headspace vials and sealed with a PTFE/silicone cap (Sigma-Aldrich). A 2 cm triphasic solid-phase microextraction (SPME) fiber consisting of polydimethylsiloxane, divinylbenzene, and carboxen (Supelco) was suspended in the headspace of the supernatant for 30 min at 37°C with 250 rpm shaking.

The SPME fiber was injected into the inlet of a Pegasus 4D (LECO Corp.) GC×GC TOFMS equipped with a rail autosampler (Gerstel), and fitted with a two-dimensional column set consisting of an Rxi-624Sil [60 m×250 μm×1.4 μm (length×internal diameter×film thickness)] first column and Stabilwax (Crossbond Carbowax polyethylene glycol; 1 m×250 μm×0.5 μm) second column (Restek). A splitless injection was performed, with the front inlet set to 270°C. The main oven containing column one was held at 35°C for 30 s, and then ramped at 3.5°C/min to a final temperature of 230°C. The secondary oven containing column two and cryogenic modulator were heated in-step with the main over with +5°C and +30°C offsets, respectively. The modulation period was set at 2.0 s, with hot- and cold-pulses alternating every 0.5 s. The transfer line temperature was set at 250°C. Mass spectra were collected over a range of 30–500 *m/z*, with an acquisition rate of 200 spectra/s, an ion source temperature of 200°C, and a detector voltage offset of +50 V.

Alignment of peaks across chromatograms was performed using the Statistical Compare feature of ChromaTOF (LECO Corp.). An inter-chromatogram mass spectral match score ≥600 (out of 1000) and maximum first and second dimension retention time deviations of 6 s and 0.15 s, respectively, were required for peak alignment. Only peaks detected at a signal-to-noise ratio of ≥50:1 in one or more chromatogram were considered for subsequent analyses. Mass spectra were compared with the National Institute of Standards and Technology (NIST) 2011 mass spectral library, and a forward match score ≥700 (out of 1000) was required for putative compound identification. When possible,

putative identifications were affirmed by comparing experimentally-determined linear retention indices (using $C_6$ to $C_{15}$ straight-chain alkanes, Sigma-Aldrich) with previously-reported values for both polar and non-polar column configurations.

Relative compound abundances (measured in total ion chromatogram (TIC)) were $\log_{10}$-transformed, mean-centered, and unit scaled prior to statistical analysis. The non-parametric Mann-Whitney U-test (*Mann and Whitney, 1947*) with Benjamini-Hochberg correction (*Benjamini and Hochberg, 1995*) was used to identify volatile metabolites that were significantly different in relative compound abundance ($p < 0.05$ after correction) between WT and $\Delta lsr2$ strains.

## Generation of an Lsr2 knockdown construct

We created the dominant negative R82A point mutant variant of Lsr2 using overlap extension PCR. Briefly, two products were generated using primers sven3225F/R82Asven3225R and R82Asven3225F/sven3225R (*Supplementary file 8*). The resulting products were gel purified, before being mixed together in a 1:1 molar ratio for subsequent stitching together and amplification. Amplification was achieved using phosphorylated oligonucleotides sven3225F and sven3225R. The resulting product was purified and cloned into pIJ82, and construct integrity was confirmed by sequencing. The R82A mutant variant was then re-amplified using NdeI3225F and PacI3225R (to remove its native promoter, which is subject to negative autoregulation), digested with *Nde*I and *Pac*I, and cloned into the same restriction enzyme sites in pIJ12551, downstream of the constitutive, highly active *ermE** promoter (*Supplementary files 7* and *8*). Construct integrity was confirmed by sequencing before being introduced by conjugation into *S. venezuelae*, *S. coelicolor*, and five different wild *Streptomyces* isolates from the Wright actinomycete collection (WAC4718, WAC5514, WAC6377, WAC7072, and WAC7520).

## Acknowledgements

We wish to thank Michael Hewak, Talha Qureshi, Dan Seale, Lilian Raphael, and Christine Pham for technical assistance, Stephen Miller for computational/data mining expertise, and Gerry Wright for access to his library of wild *Streptomyces* species. We are also grateful to Mervyn Bibb, Mark Buttner and Govind Chandra (John Innes Centre), Ben Evans (McMaster University), and Will Navarre and Jun Liu (University of Toronto) for helpful discussions. This work was supported by Cystic Fibrosis Canada (to JN and MAE), the Canadian Institutes of Health Research (to MAE) and the Boris Family Foundation (to MAE). EJG was supported by a Department of Foreign Affairs and International Trade fellowship, and the Drug Safety and Efficacy Cross-disciplinary Training program from the Canadian Institutes of Health Research.

## Additional information

### Funding

| Funder | Author |
| --- | --- |
| Cystic Fibrosis Canada | Justin R Nodwell<br>Marie A Elliot |
| Boris Family Foundation | Marie A Elliot |
| Canadian Institutes of Health Research | Marie A Elliot |

The funders had no role in study design, data collection and interpretation, or the decision to submit the work for publication.

### Author contributions

Emma J Gehrke, Conceptualization, Formal analysis, Validation, Investigation, Visualization, Methodology, Writing—original draft, Writing—review and editing; Xiafei Zhang, Andrew R Johnson, Christiaan A Rees, Formal analysis, Validation, Investigation, Visualization, Writing—original draft, Writing—review and editing; Sheila M Pimentel-Elardo, Formal analysis, Validation, Investigation, Visualization; Stephanie E Jones, Hindra, Formal analysis, Investigation, Visualization,

Writing—review and editing; Sebastian S Gehrke, Formal analysis, Investigation; Sonya Turvey, Formal analysis, Investigation, Visualization; Suzanne Boursalie, Formal analysis, Validation, Investigation; Jane E Hill, Formal analysis, Supervision, Writing—original draft; Erin E Carlson, Formal analysis, Supervision, Writing—original draft, Writing—review and editing; Justin R Nodwell, Formal analysis, Supervision, Writing—review and editing; Marie A Elliot, Conceptualization, Formal analysis, Supervision, Funding acquisition, Visualization, Writing—original draft, Project administration, Writing—review and editing

### Author ORCIDs
Hindra (iD) https://orcid.org/0000-0002-8411-8123
Marie A Elliot (iD) https://orcid.org/0000-0001-6546-5835

### Decision letter and Author response
Decision letter https://doi.org/10.7554/eLife.47691.033
Author response https://doi.org/10.7554/eLife.47691.034

## Additional files

### Supplementary files
• Supplementary file 1. Differentially expressed genes in an *lsr2* mutant, relative to wild type (*q*value <0.01;>4 fold change).
DOI: https://doi.org/10.7554/eLife.47691.021

• Supplementary file 2. Transcript levels of known global antibiotic regulatory genes.
DOI: https://doi.org/10.7554/eLife.47691.022

• Supplementary file 3. ChIP-seq identified Lsr2-FLAG-bound sequences.
DOI: https://doi.org/10.7554/eLife.47691.023

• Supplementary file 4. (**A**) Number of core binding sites and AT-rich 20 nt stretches within Lsr2 binding sequences. (**B**) Start and end positions within the chromosome, for AT-rich 'core' sequences (5 of 6 A/T residues). (**C**) Position and sequence of AT-rich 20 nt stretches. (**D**) Number of AT-rich core and 20-nt stretches in random 500 bp and 1000 bp sequences
DOI: https://doi.org/10.7554/eLife.47691.024

• Supplementary file 5. Highly abundant (intensities $> 10^5$) unique or significantly upregulated (>5 fold) compounds.
DOI: https://doi.org/10.7554/eLife.47691.025

• Supplementary file 6. Putative identification of the most abundant, differentially expressed volatile compounds by wild type and Δ*lsr2* strains of *S. venezuelae*.
DOI: https://doi.org/10.7554/eLife.47691.026

• Supplementary file 7. Bacterial strains, plasmid and cosmids used in this work.
DOI: https://doi.org/10.7554/eLife.47691.027

• Supplementary file 8. Oligonucleotides and synthetic DNA used in this study.
DOI: https://doi.org/10.7554/eLife.47691.028

• Transparent reporting form
DOI: https://doi.org/10.7554/eLife.47691.029

### Data availability
Sequencing data (RNA-seq and ChIP-seq) have been deposited in GEO under the accession code GSE115439.

The following dataset was generated:

| Author(s) | Year | Dataset title | Dataset URL | Database and Identifier |
| --- | --- | --- | --- | --- |
| Gehrke E, Elliot M | 2018 | Lsr2 binding sites and gene expression of wild type and lsr2 mutants of Streptomyces venezuelae during late | https://www.ncbi.nlm.nih.gov/geo/query/acc.cgi?acc=GSE115439 | NCBI Gene Expression Omnibus, GSE115439 |

development in liquid culture

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
