## [Decision Letter]

[Editors’ note: a previous version of this study was rejected after peer review, but the authors submitted for reconsideration. The first decision letter after peer review is shown below.]

Thank you for submitting your work entitled "Repressing cryptic specialized metabolism in *Streptomyces* by the xenogeneic silencer Lsr2" for consideration by *eLife*. Your article has been reviewed by three peer reviewers, one of whom is a member of our Board of Reviewing Editors, and the evaluation has been overseen by a Senior Editor. The reviewers have opted to remain anonymous.

Our decision has been reached after consultation between the reviewers. Based on these discussions and the individual reviews below, we regret to inform you that your work will not be considered further for publication in *eLife*.

The reviewers were all enthusiastic about the general topic and the identification of Lsr2 as a potential regulator of many, normally silent gene clusters that are responsible for producing a variety of specialized metabolites in *Streptomyces*. The identification of a dominant negative variant of Lsr2 that could potentially be used throughout this genus to induce the production of useful small molecules was also met with excitement. However, the reviewers also had concerns about the novelty of finding a NAP that globally controls such gene clusters given prior work on HU and BldM. There were also significant concerns raised about the depth of the study in terms of elucidating the mechanism(s) by which Lsr2 regulates various gene clusters and even whether it functions directly or indirectly at the loci of interest. In discussion, the reviewers agreed that the paper was not suitable for *eLife* as it stands now, but potentially could be, with significant revision and addition of mechanistic depth (as detailed in the individual reviews below). That said, the reviewers also recognized that such work is quite involved and could take a considerable amount of time, in which case it may be best to pursue publication elsewhere; the reviewers felt that with more modest revision, e.g. improvement of the statistical treatment of the global data-sets, the paper could be ready for publication in another journal in short order. In that vein, you may want to consider the peer review transfer process noted below.

Reviewer #1:

This manuscript seeks to characterize the role of a nucleoid-associated protein, Lsr2 from *Streptomyces venezuelae*, as a regulator of xenogenic DNA and specialized metabolic gene clusters. The authors use RNA-seq to determine how a deletion in Lsr2 in *Streptomyces venezuelae* impacts gene expression genome-wide, and they try to correlate this to DNA-binding sites of Lsr2 identified using ChIP-seq. Deletion of Lsr2 leads to widespread differential gene expression (primarily upregulation) of many specialized metabolic cluster genes, with some possibly directly regulated. Using a dominant negative form of Lsr2, the authors demonstrate that disruption of native Lsr2 in various isolates of *Streptomyces* potentially leads to production of previously-silenced specialized metabolites. The authors propose that this may be a new method for producing novel metabolites not normally made in laboratory conditions. Although generally clear and easy to follow, the genome-wide studies presented needed better quantification and analysis to substantiate some of the claims made. I was also somewhat disappointed by the lack of mechanistic insight into how Lsr2 associates with certain regions of DNA and how it regulates gene expression. The primary advance seems to be the demonstration that deleting Lsr2 leads to expression of specialized metabolic gene clusters and their consequent production. It wasn't clear to me whether this was a major conceptual advance for the *Streptomyces* field.

1) Subsection “Lsr2 represses the expression of horizontally acquired and specialized metabolic genes in *S. venezuelae*”, second paragraph: The authors argue that specialized metabolism genes are disproportionately affected in the *lsr2* mutant. Are there data for another NAP to compare against to support this claim? Maybe all NAPs yield such changes? Also, proper statistical testing is needed to assess whether specialized metabolic gene clusters are really disproportionately affected.

2) Nearly 75% of the sites Lsr2 is associated with by ChIP-seq don't show changes in expression – why not?

3) The authors state that there is a "good correlation" between Lsr2 binding and AT-content, but this conclusion wasn't substantiated rigorously or backed up with a quantitative analysis of the correlation. And the overall GC content of Lsr2 bound sites was only slightly less than that of the genome as a whole. So although the framework being applied here is H-NS, it seems like Lsr2 might not behave as "cleanly" with respect to binding AT-rich DNA. In the same vein, could AT-content be more systematically tested by EMSA?

4) The mechanism(s) by which Lsr2 regulates specialized metabolism gene cluster expression is not well characterized. It is suggested that for some it might bind and repress a regulatory gene, but this hypothesis is not tested, e.g. if the presumed Lsr2 binding site near a regulatory gene is mutated, does the corresponding metabolic gene cluster get upregulated?

5) Lsr2 binds near 14 specialized metabolism gene clusters. But 21 had expression changes. What explains the others? And is it true that the 14 are a subset of the 21?

6) Does Lsr2 bind to more recently acquired gene clusters? Phylogenetic studies could have substantially elevated the impact of the paper by directly addressing the hypothesis that Lsr2 binds and regulates foreign DNA. On a related note: is the idea that something of benefit over time loses Lsr2 regulation to become expressed and useful to the host? If so, is there any evidence that this is happening?

7) Is there a condition in which Lsr2 mutants show a disadvantage relative to the wild-type because they express "foreign" DNA and perhaps some of the gene clusters of interest here?

Reviewer #2:

Gehrke et al., 2018 discovered the nucleoid-associated protein Lsr2 in *Streptomyces venezuelae* and proposed that Lsr2 directly represses the expression of genes in the majority of specialized metabolic clusters in this antibiotic-producing organism. The authors exploited these knowledge to manipulate the Lsr2 activity in the model *Streptomyces* species as well as uncharacterized isolates to improve the production of new secondary metabolites. The work was done rigorously and most of the claims were experimentally validated, however the novelty of this work might be limited. Lsr2 was first discovered and characterized in Mycobacterium, and was shown to be a transcriptional repressor. Gehrke et al., 2018 has not gone further in understanding the molecular mechanism of this repressor than in the previous works in Mycobacterium. Furthermore, deletions of global transcription factors/nucleoid-associated proteins in *Streptomyces* spp. very often affects the expression of biosynthetic clusters. Some of the examples are: Salerno et al., 2009: a deletion of an HU-encoding gene affected *Streptomyces* spore pigmentation, Som et al., 2017: a global transcription regulator MtrA controls the expression of the actinorhodin and undecylprodigiosin biosynthetic clusters in *Streptomyces*, or Thanapipatsiri et al., 2016: BldM, a transcriptional regulator of *Streptomyces* development, controls the expression of the venemycin biosynthetic cluster.

Reviewer #3:

The paper by Gehrke and colleagues describes the effects of loss of Lsr2 activity on secondary metabolite production in *Streptomyces venezuelae*, the organism that produces chloramphenicol. The loss of Lsr2 not only results in dysregulation of known metabolites but also the production of novel ones. The paper also maps the binding sites of Lsr2 in the genome and reports the membership of its transcriptome. The findings are novel and will be of interest in both the basic science and the biotechnology spheres. However, there is scope to improve the scholarship in the article and to address anomalies around some of the data and to improve the presentation of some of the data.

*Streptomyces venezuelae* expresses both Lsr2 and LsrL. Although the authors study mutants deficient in LsrL expression, this protein is absent from the Introduction and is not mentioned in the Abstract (it first appears in the Results subsection “Lsr2 and Lsr2-like (LsrL) in *Streptomyces venezuelae*”). After Figure 1 (and Figure 1—figure supplement 1) it disappears from the paper, not even appearing in the Discussion. Do the authors consider LsrL to be the counterpart of StpA, a paralogue of H-NS that is expressed in *E. coli* and related Gram-negative bacteria? Given that they have invested effort in investigating LsrL, it would be useful to give some thought to its physiological role. Like the LsrL-deficient mutant in this study, stpA knockout mutants of Gram-negative bacteria have few phenotypes, but they do have some: see PMID: 11029695 and PMID: 19843227. This point seems to deserve some consideration, otherwise why include the LsrL data?

When considering the impact of Lsr2 on gene expression in *S. venezuelae*, do the authors consider it as acting exclusively at the level of transcription silencing? H-NS and its paralogues have excellent RNA binding activity and well-characterised effects on translation. This point also has relevance for the LsrL/StpA issue that was raised above. What is known about Lsr2 and posttranscriptional control in *Streptomyces* (and in Mycobacteria)? Some commentary on this possibility would deepen the scholarly aspect of the study.

The distribution of Lsr2 binding sites along the linear chromosome shown in Figure 2 seems to be non-random. There is dense clustering of the sites in the mid-section with noticeable thinning at the left and right portions. Is this significant, perhaps in terms of chromosome organisation?

There is a presentational difficulty in panels B, C and D of Figure 4. Although the peaks associated with the *lsr2* mutants are easy to see, the other data are much less discernable. For example, the WT intensity of chloramphenicol (shown by a red trace) in panel B is only visible around 6.25 minutes. Also, the inconsistency in the use of colour between panels doesn't help the reader to follow the story: WT is represented by red (panel B), blue (C) and green (D). Can these panels be re-scaled to make the messages easier to appreciate?

The use of the dominant negative Lsr2* protein (Figures 5 and Figure 6—figure supplement 1) is an interesting approach. Do the authors consider Lsr2* as acting on Lsr2 directly by forming defective dimers or defective higher oligomers, by analogy with H-NS and its protein-protein interactions? The DNA binding assay shown in Figure 6—figure supplement 1 is not wholly convincing because the free probe is only barely restored at the highest concentration of Lsr2* used and the shifted complex is still obvious at that Lsr2* concentration (representing a 4:1 ratio of Lsr2* to Lsr2). Are these protein concentrations in the physiological range?

In Figure 5B, the presence of the empty vector is not neutral: the colony morphology in the top-right quadrant of the figure is completely different to that shown by the WT in the top left quadrant. This deserves some discussion. The plasmid is described as an 'integrating' vector, so presumably it enters the chromosome through a single crossover event and remains in place in the strains expressing Lsr2*? What are the implications for the interpretation of the data from the knockdown experiments if the vector itself is altering the physiology of the bacterium (and so is not truly a 'negative' control)?

---

## [Author Response]

[Editors’ note: the author responses to the first round of peer review follow.]Reviewer #1:[…] Although generally clear and easy to follow, the genome-wide studies presented needed better quantification and analysis to substantiate some of the claims made. I was also somewhat disappointed by the lack of mechanistic insight into how Lsr2 associates with certain regions of DNA and how it regulates gene expression.

As described below, we have performed additional experiments and conducted additional analyses, to better support our genome-wide studies, and to provide greater insight into the mechanism by which Lsr2 associates with DNA and impacts gene expression. In particular, we have:

i) Investigated the distribution of Lsr2 binding sites, and correlated binding in the arm regions of the chromosome, with greater transcriptional effects.

ii) Correlated direct transcriptional effects with Lsr2 binding site size (larger) and relative AT-content (higher).

iii) Demonstrated that Lsr2 from *S. venezuelae* preferentially associates with AT-rich sequences both in vitro and in vivo, and showed that Lsr2 binding to these AT-rich sequences can directly influence the transcription of the flanking genes.

iv) Determined that Lsr2 does not solely associate with newly acquired biosynthetic clusters, and thus functions as more than just a genome sentinel in shutting down the expression of heterologously acquired sequences and in controlling specialized metabolism.

v) Expanded our analyses of LsrL, including identifying a regulatory connection with Lsr2, and undertaking a broad metabolomic profiling of both an *lsrL* and double *lsr2/lsrL* mutant strain.

The primary advance seems to be the demonstration that deleting Lsr2 leads to expression of specialized metabolic gene clusters and their consequent production. It wasn't clear to me whether this was a major conceptual advance for the Streptomyces field.

There is tremendous interest in identifying ways to stimulate cryptic specialized metabolites in the *Streptomyces*, and other metabolically rich bacteria and fungi, as evidenced by the many reviews in this area in recent years:

1) Davies, J. (2011) How to discover new antibiotics: harvesting the parvome. Curr. Opin, Chem. Biol. 15: 5-10.

2) Antoraz, S., Santamaria, R.I., Diaz, M., Sanz, D., Rodrigues, H. (2015) Toward a new focus in antibiotic and drug discovery from the Streptomyces arsenal. Front. Microbiol. 6:461.

3) Katz, L., Baltz, R.H. (2016) Natural product discovery: past, present, and future. J. Ind. Microbiol. Biotechnol. 43: 155-176.

4) Onaka, 2017.

5) Mao, D., Okada, B.K., Wu, Y., Xu, F., Seyedsayamdost, M.R. (2018) Recent advances in activating silent biosynthetic gene clusters in bacteria. Curr. Opin. Microbiol. 45: 156-163.

Importantly, why so many of these clusters are not expressed in laboratory environments has been a complete mystery, particularly for the *Streptomyces*. With Lsr2, we provide a mechanistic reason for their repression – and with it, an avenue to promote the expression of these clusters. Furthermore, we show that Lsr2 controls the expression of these clusters independently of any characterized global regulators. This means that Lsr2 represents an important new level in the regulatory hierarchy governing specialized metabolism in the streptomycetes.

We have modified our manuscript throughout, to ensure that the importance of this study is clear to readers.

1) Subsection “Lsr2 represses the expression of horizontally acquired and specialized metabolic genes in S. venezuelae”, second paragraph: The authors argue that specialized metabolism genes are disproportionately affected in the lsr2 mutant. Are there data for another NAP to compare against to support this claim? Maybe all NAPs yield such changes? Also, proper statistical testing is needed to assess whether specialized metabolic gene clusters are really disproportionately affected.

We agree with the reviewer that many NAPs may indeed impact specialized metabolism. However, there is little in the way of published data that either support or refute this possibility, as few equivalent genome-wide analyses are available for other NAPs in the streptomycetes. Gene deletion experiments have been performed for sIHF (the functional equivalent of the *E. coli* IHF protein) and for HU in *Streptomyces coelicolor*. However, no ChIP or transcriptional analyses (RNA-seq or microarray assays) have been published for either of these proteins, and consequently their binding sites remain elusive, as do their genome-wide effects on transcription. Phenotypic investigations into the effects of their gene deletions focussed more heavily on the developmental consequences stemming from deletion, where sporulation was profoundly affected in each case; this has not been observed for Lsr2.

Recent work published by Bush et al., 2019, has also suggested that BldC functions as a nucleoid associated protein. Notably, this protein preferentially impacted the expression of developmental genes, not specialized metabolic genes.

Collectively, this would suggest that not all nucleoid-associated proteins have the same effect as Lsr2.

Regarding the disproportionate impact of *lsr2* deletion on specialized metabolic clusters, we have now included the proportion of specialized metabolic cluster genes that are impacted, relative to all other genes. The difference is striking: ~15% (155/1031) of genes in specialized metabolic clusters show significantly altered transcription profiles, compared with 4.9% of non-specialized metabolic cluster genes (329/6680). This information can be found in the second paragraph of the subsection “Lsr2 represses the expression of horizontally acquired and specialized metabolic genes in *S. venezuelae*”.

2) Nearly 75% of the sites Lsr2 is associated with by ChIP-seq don't show changes in expression – why not?

There are many possible reasons to explain this observation. A likely explanation, given the chromosome architectural role ascribed to Lsr2-related proteins, is that these Lsr2 binding sites may have architectural functions as opposed to regulatory ones (although at a gross chromosomal level, we did not detect any significant changes in chromosome compaction in the *lsr2* mutant, relative to the wild type strain). We have now added this possibility into our Discussion in the second paragraph of the subsection “Mechanism of Lsr2-mediated repression”. We have also compared the characteristics of those sites associated with expression changes, with those that have no effect, and have identified a number of features that are more commonly associated with transcriptional changes (larger binding sites; greater AT-content). This information can be found in the second and third paragraphs of the subsection “Trends in Lsr2 binding and regulatory control” and in new Figure 2B and 2C.

3) The authors state that there is a "good correlation" between Lsr2 binding and AT-content, but this conclusion wasn't substantiated rigorously or backed up with a quantitative analysis of the correlation. And the overall GC content of Lsr2 bound sites was only slightly less than that of the genome as a whole. So although the framework being applied here is H-NS, it seems like Lsr2 might not behave as "cleanly" with respect to binding AT-rich DNA. In the same vein, could AT-content be more systematically tested by EMSA?

We have undertaken a more comprehensive analysis of Lsr2 binding site composition, relative to that of the general chromosome, with respect to AT-rich ‘core’ binding sites (5 out of 6 AT residues), and AT-rich stretches of 20 bp (>50% AT), and have shown that while not all sequences with these features are Lsr2 targets, the vast majority of Lsr2 targets possess these sequences.

We have also systematically tested Lsr2 binding to a sequence modified to have a range of GC contents (58%-70%), both in vitroand in vivo. We found that Lsr2 binding is enhanced with increasing ATcontent, with little binding observed when the probe was 70% GC (new Figure 3—figure supplements 3 and 4). in vivo, we altered the AT-content of previously defined binding sites (the same sequences tested in the in vitroEMSA experiments described above), and observed greater Lsr2 binding to the AT-rich sequences than the GC-rich ones (new Figure 3—figure supplement 3). We also observed an inverse correlation between flanking gene expression and Lsr2 binding: as less Lsr2 bound, more transcription was observed, confirming that Lsr2 binding to AT-rich sequences served to repress gene expression.

These experiments are all detailed in a new Results section (subsection “Trends in Lsr2 binding and regulatory control”).

4) The mechanism(s) by which Lsr2 regulates specialized metabolism gene cluster expression is not well characterized. It is suggested that for some it might bind and repress a regulatory gene, but this hypothesis is not tested, e.g. if the presumed Lsr2 binding site near a regulatory gene is mutated, does the corresponding metabolic gene cluster get upregulated?

Many of the Lsr2 binding sites associated with regulatory genes overlap or are within the coding sequences of their associated regulatory genes, making it challenging to mutate these sites without changing the coding sequence of the associated gene, or significantly altering the codon usage of these genes. However, we identified one candidate binding site that was ideally positioned for testing this hypothesis, in that it was both upstream of a regulatory gene (*sven_5106*), and the expression of this regulatory gene was altered in an *lsr2* mutant background.

We introduced this region into a plasmid vector, and compared the expression of genes flanking the Lsr2 binding site bearing either the native (AT-rich) sequence, or more GC-rich variants, where the ChIP-identified AT-rich binding core sequence had been altered (taking care not to mutate either promoter sequence within this region). These altered sequences (and the corresponding wild type) were the ones described in point 3, and were tested in vitroand in vivofor Lsr2 binding (binding was strongest to the wild type, or most AT-rich, and was weakest to the sequence with the highest GCcontent).

We found that increasing GC-content (and decreasing Lsr2-binding) led to increased expression of the flanking genes, as determined using RT-qPCR. These data are presented in Figure 3—figure supplements 3 and 4, and are described in the last paragraph of the subsection “Trends in Lsr2 binding and regulatory control”.

5) Lsr2 binds near 14 specialized metabolism gene clusters. But 21 had expression changes. What explains the others? And is it true that the 14 are a subset of the 21?

Lsr2 binds within 17 specialized metabolic gene clusters, and there are expression changes associated with genes in 14 of these clusters. These 14 are indeed a subset of the 21 clusters having changed expression patterns – please see Table 1 for a complete summary of this information. For the 7 clusters that had transcriptional changes but no direct Lsr2 binding, we predict that these transcriptional effects are indirect. These may be mediated by other transcription factors under the control of Lsr2, or by alterations in the local DNA architecture stemming from the loss of Lsr2 binding. We have now elaborated upon these possibilities in our Discussion (subsection “Mechanism of Lsr2-mediated repression”).

6) Does Lsr2 bind to more recently acquired gene clusters? Phylogenetic studies could have substantially elevated the impact of the paper by directly addressing the hypothesis that Lsr2 binds and regulates foreign DNA. On a related note: is the idea that something of benefit over time loses Lsr2 regulation to become expressed and useful to the host? If so, is there any evidence that this is happening?

We have conducted phylogenetic analyses of each cluster, to determine whether Lsr2 preferentially associated with newly acquired (not conserved/unique to *S. venezuelae*) clusters, and to our surprise, found that while Lsr2 often bound to poorly conserved clusters, it also associated with (and affected the expression of) several of the most conserved clusters (see new Figure 4). This suggested that Lsr2’s function in controlling specialized metabolism extends beyond simply repressing the expression of foreign/newly acquired genes, and indicates that it also represents an important level in the regulatory cascades governing specialized metabolism in general. We have added in a new section addressing this point within our Results (subsection “Lsr2 activity is not specific for newly acquired biosynthetic clusters”, first paragraph), and have expanded upon it in our Discussion (subsection " Role for Lsr2 repression”).

7) Is there a condition in which Lsr2 mutants show a disadvantage relative to the wild-type because they express "foreign" DNA and perhaps some of the gene clusters of interest here?

While we have not conducted a comprehensive survey of growth conditions, we have yet to determine conditions in which loss of Lsr2 is deleterious to the growth of *S. venezuelae*, apart from the delay in reproductive growth observed on and in MYM medium. We have now compared the growth of wild type *S. venezuelae* and the *lsr2* mutant in MYM, R2 and minimal media, and in all cases, the *lsr2* mutant grew similarly to the wild type (see Figure 1 in the manuscript, and Author response image 1). We also compared the growth of the wild isolates carrying the empty plasmid, with those containing the Lsr2 knock-down construct, and again, observed similar growth profiles in all cases. These results are now included in Figure 6—figure supplement 2.

We would note, however, that there is some evidence that *lsr2* is an essential gene in some streptomycetes (Najah et al., 2019). This also seems to be the case for *Streptomyces coelicolor* in our hands (unpublished data). Whether this is because of uncontrolled metabolite expression, or some other reason, is not known at this stage. However, our knock-down construct was effectively tolerated in *S. coelicolor*, and thus in strains where Lsr2 is required for viability, this construct may help to circumvent the deleterious consequences associated with complete loss of *lsr2,* while at the same time providing access to new metabolites.

Reviewer #2:[…] The work was done rigorously and most of the claims were experimentally validated, however the novelty of this work might be limited. Lsr2 was first discovered and characterized in Mycobacterium, and was shown to be a transcriptional repressor. Gehrke et al., 2018 has not gone further in understanding the molecular mechanism of this repressor than in the previous works in Mycobacterium.

To further our understanding of the molecular mechanisms underlying Lsr2 function in the streptomycetes, we have now undertaken a more comprehensive analysis of our data, that including integrating our ChIP binding data with our transcriptional analyses. We have also systematically probed the AT-requirements for both Lsr2 binding (in vitroand in vivo), and the effect of this on transcription in vivo. Specifically, and as summarized above, we have:

i) Investigated the distribution of Lsr2 binding sites, and correlated binding in the arm regions of the chromosome, with greater transcriptional effects.

ii) Correlated direct transcriptional effects with Lsr2 binding site size (larger) and relative ATcontent (higher).

iii) Demonstrated that Lsr2 from *S. venezuelae* preferentially associates with AT-rich sequences both in vitro and in vivo, and showed that Lsr2 binding to these AT-rich sequences can directly influence the transcription of the flanking genes.

iv) Determined that Lsr2 does not solely associate with newly acquired biosynthetic clusters, and thus functions as more than just a genome sentinel in shutting down the expression of heterologously acquired sequences and in controlling specialized metabolism.

v) Expanded our analyses of LsrL, including identifying a regulatory connection with Lsr2, and undertaking a broad metabolomic profiling of both an *lsrL* and double *lsr2/lsrL* mutant strain.

In addition to expanding on our mechanistic understanding of Lsr2 activity, our work has revealed Lsr2 to be one of the major reasons that specialized metabolic clusters are not expressed in laboratory environments – the reasons for this have been entirely unknown to this point. Our findings therefore will provide the field with strategies to stimulate the expression of these clusters (deleting *lsr2* or down-regulating Lsr2 activity). Intriguingly, Lsr2 also controlled the expression of a number of well conserved clusters that are known to be expressed under specific conditions (*e.g.* the siderophore desferrioxamine under low iron conditions), suggesting that Lsr2 occupies a unique niche in the regulatory hierarchy governing specialized metabolism.

Furthermore, deletions of global transcription factors/nucleoid-associated proteins in Streptomyces spp. very often affects the expression of biosynthetic clusters. Some of the examples are: Salerno et al., 2009: a deletion of an HU-encoding gene affected Streptomyces spore pigmentation, Som et al., 2017: a global transcription regulator MtrA controls the expression of the actinorhodin and undecylprodigiosin biosynthetic clusters in Streptomyces, or Thanapipatsiri et al., 2016: BldM, a transcriptional regulator of Streptomyces development, controls the expression of the venemycin biosynthetic cluster.

There have been many very nice studies into the effects of DNA binding proteins on both *Streptomyces* development and specialized metabolism, and we agree that many global transcription factors and nucleoid-associated proteins affect specialized metabolism. Considering the specific protein examples raised by the reviewer:

HU: in Salerno et al., a single specialized metabolite was identified as being affected by the loss of this protein (spore pigment), and there are no data available for its binding sites within the chromosome

MtrA: is an actinobacterial-specific two-component regulator whose activity affects characterized clusters in *S. coelicolor* and *S. venezuelae*; its effect on cryptic clusters is entirely unknown at this point.

BldM: is best known as a developmental regulator, and its effect on specialized metabolism in *S. venezuelae* was examined only at the transcriptional level. It is not clear if the BldM effect on venemycin was a direct or an indirect regulatory effect, nor whether this strain has additional mutations that may have been responsible for the effects observed (no complementation of the increased venemycin phenotype was performed).

Recent work on BldC (Bush et al., 2019) has suggested that BldC may function as a nucleoid-associated protein; however, its effect is largely confined to development, and not specialized metabolism. This would suggest that not all nucleoid associated proteins affect specialized metabolism in a similar way as Lsr2.

Here: Lsr2 is shown to directly regulate nearly half of all specialized metabolic clusters in *S. venezuelae*, and to directly or indirectly affect the transcription of 2/3 of these clusters. Notably, many of these are cryptic clusters that are not expressed in the wild type strain (unlike the situation with MtrA, based on publications to date). Such comprehensive control of specialized metabolism has not been demonstrated for any regulator in the streptomycetes to date.

Reviewer #3:The paper by Gehrke and colleagues describes the effects of loss of Lsr2 activity on secondary metabolite production in Streptomyces venezuelae, the organism that produces chloramphenicol. The loss of Lsr2 not only results in dysregulation of known metabolites but also the production of novel ones. The paper also maps the binding sites of Lsr2 in the genome and reports the membership of its transcriptome. The findings are novel and will be of interest in both the basic science and the biotechnology spheres. However, there is scope to improve the scholarship in the article and to address anomalies around some of the data and to improve the presentation of some of the data.

We have worked to improve the scholarship through more systematic and comprehensive analyses and discussion, have performed additional experiments to help resolve any anomalies in the data, and have improved the data presentation, as detailed above and below.

Streptomyces venezuelae expresses both Lsr2 and LsrL. Although the authors study mutants deficient in LsrL expression, this protein is absent from the Introduction and is not mentioned in the Abstract (it first appears in the Results subsection “Lsr2 and Lsr2-like (LsrL) in *Streptomyces venezuelae*”). After Figure 1 (and Figure 1—figure supplement 1) it disappears from the paper, not even appearing in the Discussion. Do the authors consider LsrL to be the counterpart of StpA, a paralogue of H-NS that is expressed in *E. coli* and related Gram-negative bacteria? Given that they have invested effort in investigating LsrL, it would be useful to give some thought to its physiological role. Like the LsrL-deficient mutant in this study, stpA knockout mutants of Gram-negative bacteria have few phenotypes, but they do have some: see PMID: 11029695 and PMID: 19843227. This point seems to deserve some consideration, otherwise why include the LsrL data?

While we do not currently have any data to support an StpA-like role for LsrL, we consider this to be an appealing possibility. We would note, however, that there is evidence for regulatory interplay between these two genes/proteins, with *lsrL* being directly repressed by Lsr2, similar to what has been observed for H-NS and StpA. We have now incorporated this observation, alongside metabolic analyses of the *lsrL* and *lsr2 lsrL* double mutant, in our Results section (subsection “Lsr2 represses the expression of horizontally acquired and specialized metabolic genesin *S. venezuelae*”, second paragraph and subsection “Deleting Lsr2 reprograms specialized metabolism and yields novel compounds”, second paragraph, plus new Figure 5—figure supplement 1), and have provided additional commentary on functional possibilities for LsrL, into our Discussion (subsection “Control of Lsr2 expression and activity”, third paragraph).

When considering the impact of Lsr2 on gene expression in S. venezuelae, do the authors consider it as acting exclusively at the level of transcription silencing? H-NS and its paralogues have excellent RNA binding activity and well-characterised effects on translation. This point also has relevance for the LsrL/StpA issue that was raised above. What is known about Lsr2 and posttranscriptional control in Streptomyces (and in Mycobacteria)? Some commentary on this possibility would deepen the scholarly aspect of the study.

We do not currently have any evidence for Lsr2 activity beyond that of a DNA-binding protein and transcriptional regulator. The possibility of broader effects (e.g. RNA stability and translation) is intriguing, and we have now added in additional discussion of these possibilities in the last paragraph of the subsection “Mechanism of Lsr2-mediated repression”. Beyond discussing possible post-transcriptional activities for Lsr2, we also now address possible mechanisms by which Lsr2 activity and expression could be controlled (subsection “Control of Lsr2 expression and activity”).

The distribution of Lsr2 binding sites along the linear chromosome shown in Figure 2 seems to be non-random. There is dense clustering of the sites in the mid-section with noticeable thinning at the left and right portions. Is this significant, perhaps in terms of chromosome organisation?

While we are not yet certain as to the significance of the Lsr2 binding distribution, we would agree that it may well have an effect on chromosome organization, and now raise this point in our Discussion (subsection “Mechanism of Lsr2-mediated repression”, second paragraph). A more thorough analysis of our data led us to determine that there is indeed clustering of Lsr2 binding in the ‘core’ region of the chromosome. Interestingly, binding within this central part of the chromosome is less often associated with transcriptional changes than binding to sites in the flanking ‘arm’ regions of the chromosome. We have added in this information as part of a new Results section (subsection “Trends in Lsr2 binding and regulatory control”).

There is a presentational difficulty in panels B, C and D of Figure 4. Although the peaks associated with the lsr2 mutants are easy to see, the other data are much less discernable. For example, the WT intensity of chloramphenicol (shown by a red trace) in panel B is only visible around 6.25 minutes. Also, the inconsistency in the use of colour between panels doesn't help the reader to follow the story: WT is represented by red (panel B), blue (C) and green (D). Can these panels be re-scaled to make the messages easier to appreciate?

We have adjusted the data presented in panels B, C and D (now Figure 5), to ensure that they are consistently coloured, and are hopefully easier for readers to interpret.

The use of the dominant negative Lsr2* protein (Figures 5 and Figure 6—figure supplement 1) is an interesting approach. Do the authors consider Lsr2* as acting on Lsr2 directly by forming defective dimers or defective higher oligomers, by analogy with H-NS and its protein-protein interactions? The DNA binding assay shown in Figure 6—figure supplement 1 is not wholly convincing because the free probe is only barely restored at the highest concentration of Lsr2* used and the shifted complex is still obvious at that Lsr2* concentration (representing a 4:1 ratio of Lsr2* to Lsr2). Are these protein concentrations in the physiological range?

We are not able to quantify the levels of Lsr2* reached within *S. venezuelae*, as we do not have antibodies to Lsr2 (our western blots for Lsr2 were conducted using a FLAG-tag-specific antibody). To evaluate the effect of Lsr2* in more physiologically relevant conditions, we assessed its impact on Lsr2 in vivo. We quantified wild type Lsr2-3×FLAG binding to a target sequence (*sven_0926*), when Lsr2* was being overexpressed, versus in the absence of Lsr2* (new Figure 5—figure supplement 2C). We found that Lsr2 binding to this sequence decreased by ~40% in the presence of Lsr2*. At this point, it is difficult to determine whether this is a result of defective dimer or defective oligomer formation.

Work from other groups is suggesting that *lsr2* can be an essential gene in some *Streptomyces* (Najah et al., 2019), so retaining some level of Lsr2 activity, as we show here, could therefore be critical in these cases.

In Figure 5B, the presence of the empty vector is not neutral: the colony morphology in the top-right quadrant of the figure is completely different to that shown by the WT in the top left quadrant. This deserves some discussion. The plasmid is described as an 'integrating' vector, so presumably it enters the chromosome through a single crossover event and remains in place in the strains expressing Lsr2*? What are the implications for the interpretation of the data from the knockdown experiments if the vector itself is altering the physiology of the bacterium (and so is not truly a 'negative' control)?

Integrating plasmid vectors in *Streptomyces* are indeed incorporated into the chromosome through a single recombination event, and are stably inherited in the absence of antibiotic selection. It is well established that the integration of these vectors can impact development and antibiotic production. Because of this, all comparisons of our Lsr2 knock-down strains were made to plasmid alone containing strains. While the plasmid-alone control may not be the ideal ‘negative’ control, it allows us to differentiate between metabolic changes resulting from plasmid integration, versus those resulting from Lsr2* knockdown activity.